# Efficient RL Training for LLMs with Experience Replay

**Charles Arnal** [* 1]   **Vivien Cabannes** [* 1]   **Taco Cohen** [1]   **Julia Kempe** [1 2]   **Remi Munos** [† 1]

## Abstract

While Experience Replay—the practice of storing rollouts and reusing them multiple times during training—is a foundational technique in general RL, it remains largely unexplored in LLM post-training due to the prevailing belief that fresh, on-policy data is essential for high performance. In this work, we challenge this assumption. We present a systematic study of replay buffers for LLM post-training, formalizing the optimal design as a trade-off between staleness-induced variance, sample diversity and the high computational cost of generation. We show that strict on-policy sampling is suboptimal when generation is expensive. Empirically, we show that a well-designed replay buffer can drastically reduce inference compute without degrading – and in some cases even improving – final model performance, while preserving policy entropy.

## 1. Introduction

Reinforcement Learning (RL) has emerged as the key driver behind the reasoning capabilities of modern Large Language Models (LLMs), enabling breakthroughs in complex tasks such as mathematics and coding (DeepSeek et al., 2025; OpenR1 et al., 2025). However, this performance comes at a prohibitive computational cost. Unlike pre-training, where data is static, RL requires the continuous generation of new training trajectories. In state-of-the-art pipelines, this inference cost often dominates the training budget, and may consume more than 80% of post-training GPU hours. Standard approaches exacerbate this issue through extreme sample inefficiency: methods like PPO or GRPO typically operate as on-policy as possible, meaning *rollouts are generated, used for a single gradient update, and immediately discarded.*

This "generate-then-discard" paradigm stands in stark con-

trast to classical Reinforcement Learning, where Experience Replay, i.e. storing and reusing past trajectories in a buffer, is a foundational tool for sample efficiency (Mnih et al., 2015; Lin, 1992). While Experience Replay is standard in sample-limited robotics or gaming environments, it has been largely overlooked in LLM training, where the prevailing consensus suggests that the performance degradation from off-policy data outweighs the computational benefits.

In this work, we challenge this consensus. We demonstrate that *discarding trajectories after a single use is computationally suboptimal.* By incorporating a replay buffer into asynchronous training pipelines, we trade a controlled increase in data off-policiness (staleness) and a decrease in data diversity for a dramatic reduction in inference costs. We formalize this trade-off through a theoretical analysis of the bias-variance decomposition in stochastic gradient descent, proving that optimal compute efficiency is achieved not by being strictly on-policy, but by balancing the freshness and diversity of data against its generating cost. Our contributions are as follows:

- **Theoretical Analysis:** We detail the implementation of replay buffers in asynchronous LLM training and provide a mathematical framework quantifying the trade-off between compute efficiency, sample diversity, and gradient bias. We derive theoretical bounds for the optimal buffer size and replay ratio, showing that as the relative cost of inference increases, the optimal strategy shifts further towards experience replay.

- **Empirical Analysis:** Through extensive experiments, we provide an in-depth analysis of how buffer hyperparameters influence the training process. We show that while aggressive reuse of samples can degrade performance, a well-sized buffer acts as a regularizer that stabilizes training and preserves model output diversity (improving pass@$k$ metrics).

- **Empirical Gains:** We validate those conclusions on larger models and show that simple, easy-to-implement buffer strategies can save *up to 40% of the compute budget while maintaining, and sometimes surpassing, the same final accuracy as the on-policy baseline*, as shown e.g. in Figure 1. We further explore how more sophisticated sampling strategies (e.g., prioritizing pos-

---

[1]FAIR at Meta [2]NYU Courant Institute and CDS. Correspondence to: Charles Arnal <charlesarnal@meta.com>.

*Proceedings of the $43^{rd}$ International Conference on Machine Learning*, Seoul, South Korea. PMLR 306, 2026. Copyright 2026 by the author(s).

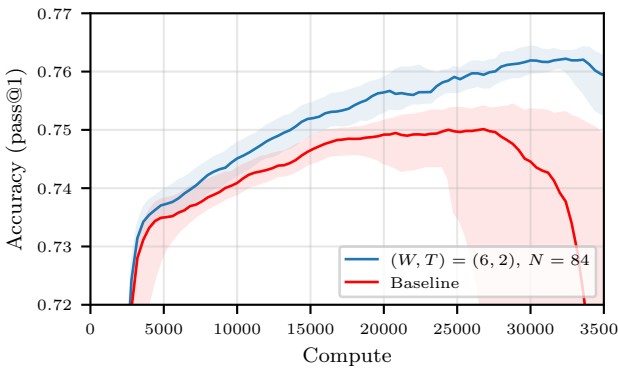

*Figure 1.* **Experience Replay improves LLM RL Training.** Accuracy on MATH as a function of compute spent when training Qwen2.5-7B on OpenR1-Math-220k for a no-buffer baseline (orange curve) and a buffer of size 84 with 6 inference worker GPUS and 2 trainer GPUs. We report the median and IQR over 10 seeds. Compute is calibrated so that a single weight update for the baseline costs 1 unit. Baseline runs display increased instability.

itive trajectories) and alternative losses can extend the stability of replay buffers, allowing for even greater efficiency gains.

Through this study, we present a straightforward approach for high-efficiency RL fine-tuning, shifting the focus from maximizing performance per step to *maximizing performance per unit of compute.*

## 2. Related Work

**Experience Replay in RL.** The use of a replay buffer is a cornerstone of deep RL, famously enabling stability and sample efficiency in algorithms like DQN (Mnih et al., 2015), Soft Actor-Critic (Haarnoja et al., 2018), and DDPG (Lillicrap et al., 2015). Techniques such as Prioritized Experience Replay (Schaul et al., 2015) and Hindsight Experience Replay (Andrychowicz et al., 2017) further optimized how agents learn from past data. Despite this rich history, modern LLM reasoning pipelines (DeepSeek et al., 2025; OpenR1 et al., 2025) have largely defaulted to on-policy training (e.g., GRPO), discarding trajectories immediately after a gradient update to avoid off-policy degradation, though, in practice, implementation constraints typically lead to some unavoidable off-policiness.

**Replay Buffers for LLMs.** In contrast to the mainstream approach, several works have re-introduced replay buffers to LLM training, though with different motivations. Wang et al. (2025) and Bartoldson et al. (2025) utilize buffers primarily to enhance exploration and final model performance, often requiring specialized loss functions or complex filtering. Similarly, Lu et al. (2025) and Zhang et al. (2025) propose dynamic sampling or multi-phase training to maximize data quality. In contrast, our work focuses strictly on *compute*

*efficiency*. We do not propose a new training paradigm to beat state-of-the-art accuracy; rather, we systematically analyze the trade-off between off-policiness and efficiency in standard asynchronous pipelines, demonstrating that simple experience replay can drastically reduce the compute budget while maintaining accuracy.

A more detailed discussion of off-policy algorithms and related theoretical works is provided in Appendix A.

## 3. Experience Replay for Off-Policy RL

We present how experience replay can be efficiently implemented in an LLM post-training pipeline and discuss the role of various hyperparameters and their impact on compute efficiency.

### 3.1. Reinforcement Learning and Replay Buffers

In modern, compute-efficient RL pipelines for LLMs, the GPUs are often split between $W$ *inference workers* and $T$ *trainers* (Noukhovitch et al., 2024; Gehring et al., 2024; Wu et al., 2025; Bartoldson et al., 2025; FAIR CodeGen Team et al., 2025). At any given time, each of the two groups maintains its own (possibly stale) copy of the model weights. The inference workers continuously generate trajectories (also called rollouts) using their set of weights, then pass them to the trainers, usually via a transfer queue. Concurrently, trainers pull trajectories from the queue, perform forward-backward passes over them and update their weights. Trajectories are discarded after having been used once (Schulman et al., 2017; Shao et al., 2024; DeepSeek et al., 2025). Every few gradient steps, the inference worker's weights are updated with the current value of the trainers' weights. This setting, which corresponds to our experimental implementation, is sometimes referred to as *asynchronous training.* *Synchronous* setups also exist (von Werra et al., 2020; Sheng et al., 2024); we discuss them in Section 4.

A *replay buffer* can be implemented as follows: instead of adding their rollouts to a queue, the inference workers add them to a list of trajectories, the replay buffer. In parallel, trainers continuously sample from this replay buffer; sampling from the buffer does not remove the sampled trajectories from it.[1] This allows for the re-using of samples, which in turn reduces the amount of overall compute needed by amortizing the cost of rollout generation, as detailed further below. Pseudo-code is provided in Appendix B.

The replay buffer can be sampled by the trainers to assemble their training batches following several strategies that pick samples based on characteristics such as their recency, their associated rewards, the norm of past gradients computed

---

[1]In our specific implementation, the buffer is sharded across trainers; see Appendix D for details.

using them, or how many times the rollout has already been sampled. One might also want to define a decay rule for the buffer, e.g. making the buffer a first-in, first-out list by keeping only the $N$ freshest samples.

### 3.2. Off-Policiness, Diversity, and Compute Efficiency

The design of the buffer and the ratio $W/T$ of inference workers to trainers directly impact three major aspects of training: the *compute efficiency*, the *degree of off-policiness*, and the *diversity of the samples*.

To illustrate these concepts, we consider throughout this subsection a buffer configuration with $T \in \{1, 2, \ldots, 7\}$ trainer GPUs, $W := 8 - T$ inference worker GPUs, and a first-in, first-out buffer (i.e. that contains the last $N$ samples generated by the inference workers). Training samples are drawn uniformly at random from the buffer at each step. We train Qwen2.5-7B (Qwen et al., 2025) model on the OpenR1-Math-220k reasoning dataset (OpenR1 et al., 2025) (see Subsection 5.1 for experimental details).

**Compute Efficiency** The compute spent on an RL training run, which we think of in terms of active GPU seconds[2], can be decomposed roughly as the sum of the *trainer compute*, spent on forward-backward passes and weight updates, and the *inference compute*, spent on generating rollouts, i.e. compute $\cong$ trainer compute + inference compute. In the asynchronous setting and without a buffer, the ratio $W/T$ of inference worker GPUs to trainer GPUs admits an optimal value $\mu$ that minimizes GPU downtime. Indeed, as a first order approximation, we can assume that the trainer compute $C$ needed for a step (including forward and backward passes and weight update) depends only on the (fixed) batch size, and not on the number of trainer GPUs. Let $\mu > 0$ be the factor such that producing a batch of rollouts of the same size costs $C \cdot \mu$ compute for the inference workers.[3] The total compute needed for each parameter update is then roughly

$$\text{compute without buffer} \approx C(1 + \mu). \tag{1}$$

In that case, the optimal ratio of inference worker GPUs to trainer GPUs, i.e. the ratio such that trainer GPUs process generated rollouts exactly at the speed at which inference worker GPUs produce them, so that neither have any downtime, is precisely $\mu$: if generating rollouts is $\mu$ times more costly than training on them, one needs $\mu$ times more inference GPUs than trainer GPUs.

By contrast, *when using a replay buffer, the inference compute is decoupled from the trainer compute*: inference work-

ers can always continuously add trajectories to the buffer from which the trainers can freely pull, independently from how many inference workers and trainers there are. As in the case without buffer, each backward pass costs $C$ trainer compute. On the other hand, the inference compute spent during a backward pass depends on the number of inference worker GPUs that are concurrently working. Hence, the total compute spent for each parameter update is roughly equal to

$$\text{total compute with buffer} \approx C(1 + W/T).$$

As reflected in this formula, when using a buffer, *increasing the number of trainers relative to the number of inference workers makes each gradient step cheaper*; intuitively, this is simply because rollouts are re-used more times on average, meaning that for a given number of optimization steps, fewer rollouts need to be generated.

We define the *compute ratio* of a buffer configuration to be

$$\gamma := \frac{1 + W/T}{1 + \mu}, \tag{2}$$

that is, the ratio of the compute cost of a parameter update with and without a buffer.

| **(W, T)** | (7,1) | (6,2) | (5,3) | (4,4) | (2,6) | (1,7) |
|---|---|---|---|---|---|---|
| $\gamma$ | 1.29 | 0.65 | 0.43 | 0.32 | 0.22 | 0.18 |

*Table 1.* $\gamma$ for various values of $(W, T)$ and an estimated $\mu = 5.28$ for Qwen2.5-7B.

**Degree of Off-Policiness** The design of the replay buffer and the ratio $(W, T)$ directly impact the off-policiness of the training distribution. We define the *off-policiness* (or staleness) of a sample used in a gradient update as the difference between the step at which the sample was created and the current step. The average off-policiness over all samples is influenced by both the size $N$ of the buffer (the larger the buffer, the greater the average off-policiness of the samples that it contains) and the ratio $W/T$ of inference worker to trainer GPUs: the more trainer GPUs there are, the faster weight updates occur and the faster samples become outdated. This can be observed on the left of Figure 2, where the distribution of off-policiness over all the samples used through a training run is represented for various pairs $(W, T)$ and buffer sizes $N$.

**Diversity of Samples** The use of a replay buffer may deteriorate training dynamics: as the same samples are reused, the training distribution seen by the policy gradient algorithm becomes less diverse, and less information regarding the true objective function is utilized. This notion of *sample diversity* arises at two distinct levels. First, the *global diversity* of samples, which we measure using the *replay ratio* of

---

[2]We explain in greater details our simplifying assumptions in Appendix D.

[3]In our experiments, we find that $\mu$ ranges from ∼4 to ∼10 depending on the model, task and implementation considered.

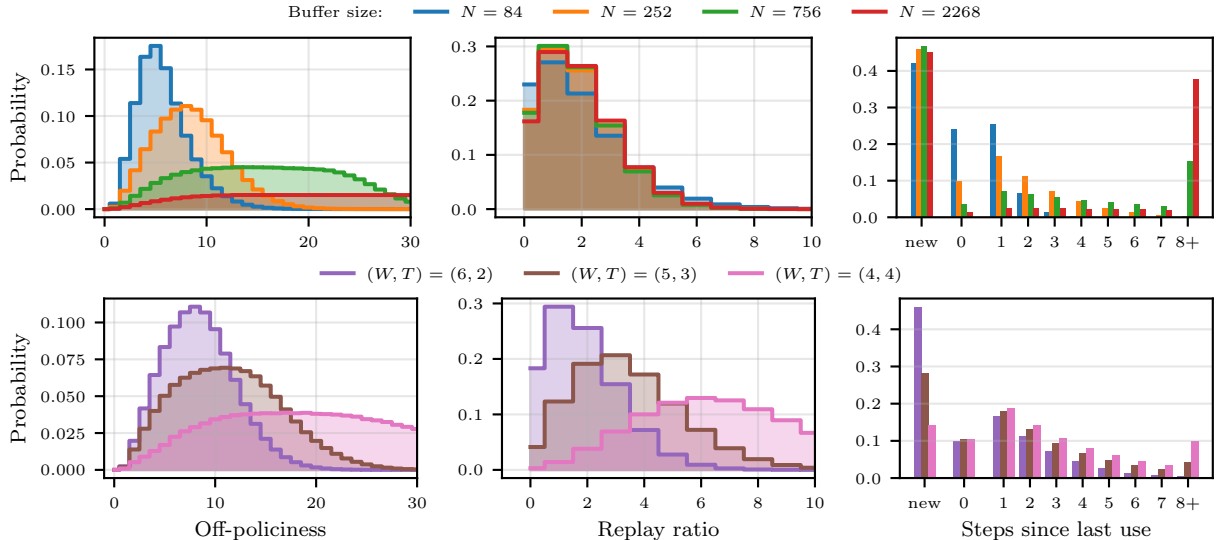

*Figure 2.* **Effect of Experimental Design on Off-Policiness and Diversity Statistics.** *Top row*: Distribution of off-policiness, replay ratio and steps-since-last-use over all samples and uses of samples during a training run for buffer size $N \in \{84, 252, 756, 2268\}$ and $(W, T) = (6, 2)$. See also Appendix D.3 for details on the steps-since-last-use metric. *Bottom row*: Same statistics for $N = 252$ and $(W, T) \in \{(6, 2), (5, 3), (4, 4)\}$. The average replay ratio is 1.78, 3.42 and 7.0 for $(W, T)$ equal to $(6, 2)$, $(5, 3)$ and $(4, 4)$ respectively.

the samples, defined as the number of times a sample has been used for a gradient step over the entire training run. The average replay ratio will be chiefly conditioned by the ratio $W/T$: the more trainer GPUs there are relative to the number of inference worker GPUs, the more passes they will do on average on each data point. This is illustrated in the middle of Figure 2.

Second, the *local diversity* of samples which is the degree to which samples are repeatedly used in close succession. We measure local diversity using the *time-since-last-use* of the samples in the current trainer batch, i.e. the number of gradient steps since the last gradient update to which they contributed. We expect a loss in local diversity to be more harmful than a loss in global diversity. At a fixed ratio $W/T$, one can trade off-policiness for local diversity: by increasing the size of the buffer, the training distribution's degree of off-policiness will increase (as discussed earlier), but the empirical training distribution will be locally more diverse: though samples are just as likely to be reused over the entire training run, they are less likely to be reused in close succession (due to the greater number of candidate samples in the buffer). This can be seen on the right side of Figure 2.

**Goal: Increased Efficiency, Preserved Accuracy** The primary motivation behind the use of a replay buffer is to save inference compute by reusing trajectories. As explained above, *each gradient step (including the required sampling) can be made computationally cheaper by letting the ratio $W/T$ decrease.* However, we have also seen that *letting the ratio $W/T$ decrease makes the training distribu-*

*tion more off-policy and less diverse.* It is usually assumed that high off-policiness and low sample diversity should be avoided (see however Tang et al. (2025); Arnal et al. (2025) and Charton & Kempe (2024)). Hence there is a *trade-off*: re-using samples from the buffer makes each gradient step cheaper, but resampling too aggressively might end up hurting the expected accuracy gain from each step. In our experiments, we explore the *efficiency/accuracy optimality curve*; in other words, *we want to maximize the accuracy achievable at a given compute cost by selecting the best buffer configuration*. To ensure our conclusions are readily applicable to production environments, we also deliberately prioritize simple implementations that require only modest departures from current SOTA pipelines.

## 4. Mathematical Analysis

While the previous section and our experiments focus on the more compute-efficient asynchronous RL setting, we choose to conduct our mathematical analysis in the conceptually simpler synchronous setting, in which the training alternates between two clearly distinct modes: a generating phase, in which new trajectories are created, and a training phase, during which a gradient descent step is performed using the new rollouts. We consider a simple first-in, first-out replay buffer: at each training step $t$, we (i) generate $R$ new rollouts using the current policy and insert them at the beginning of a buffer of capacity $N$ (evicting the oldest samples), and (ii) sample a minibatch of size $B$ uniformly from the buffer to

form a gradient update

$$\theta_{t+1} = \theta_t - \eta\, g_t, \qquad g_t = \frac{1}{B} \sum_{j=1}^{B} G(\theta_t, z_{t,i_j}).$$

Here, $\theta$ denotes the policy parameters, $z_{t,i}$ denotes the $i$-th element of the buffer at step $t$, $i_j$ the $j$-th sampled index, and $G(\theta, z)$ denotes the corresponding gradient estimate of $\nabla F(\theta)$, where $F$ is the objective we wish to minimize. The compute cost of such an update, expressed in arbitrary units, is given by $c = B + \mu R$, where $\mu$ denotes the compute cost ratio between a forward-backward pass and one rollout generation, matching the definition above.

The goal of our theoretical analysis is to characterize how the design of the replay buffer affects learning efficiency from a theoretical standpoint. We adopt the classical non-convex stochastic optimization framework and study the convergence of the training dynamics toward stationary points, as measured by the decay of the expected squared norm of the gradient. Unless stated otherwise, all norms are Euclidean.

**Assumption 4.1** (Target Smoothness). The function $F$ is non-negative, differentiable, and $L$-smooth, i.e.

$$\forall\, x, y \quad \|\nabla F(y) - \nabla F(x)\| \le L \|y - x\|.$$

Let $\mathcal{F}_t$ represent the information available from the parameter iterates up to time $t$, i.e. the $\sigma$-field associated to the sequence $(\theta_s)_{s \le t}$. Define the per-sample and minibatch gradient noises by

$$\varepsilon_{t,i} = G(\theta_t, z_{t,i}) - \nabla F(\theta_t), \quad \text{and} \quad \varepsilon_t = \frac{1}{B} \sum_{j=1}^{B} \varepsilon_{t,i_j}.$$

In contrast to usual SGD analysis, experience replay introduces a bias in the gradient estimate through the correlation introduced by the buffer, even with importance ratio correction.[4] We expect this bias to be larger when trajectories presently in the buffer have had a strong influence on the subsequent updates leading to the current parameter $\theta_t$, and to be small when the parameters have moved little over the time span covered by the buffer. This intuition motivates the following assumption, discussed further in Appendix C.

**Assumption 4.2** (Bias). There exists a constant $\kappa \ge 0$ such that for all $(t, i)$,

$$\|\mathbb{E}[\varepsilon_{t,i} \mid \mathcal{F}_t]\| \le \kappa \|\theta_t - \theta_{t_i}\|,$$

---

[4]While importance sampling corrects the marginal distribution mismatch between $\pi_{\theta_t}$ and $\pi_{\theta_{t-\tau}}$, experience replay forces us to reason about previous distributions conditioned on the current parameters, i.e. the distribution $\pi_{\theta_{t-\tau}}(\cdot \mid \theta_t)$ at time $\theta_{t-\tau}$ conditioned on the fact that the training trajectory that followed (which was influenced by the samples drawn at $\theta_{t-\tau}$ through the policy gradient algorithm) ended up at $\theta_t$. The distribution $\pi_{\theta_{t-\tau}}(\cdot \mid \theta_t)$ is typically not computable.

where $t_i = t + 1 - \lceil i/R \rceil$ is the time at which the $i$-th element of the buffer was added to the buffer.

The variance of our gradient estimates depends on both the per-sample variance, and the correlation between different samples drawn within the same minibatch. The per-sample variance typically increases with off-policiness, reflecting the growing variance of importance ratio as off-policiness increases. In addition, samples within a batch can be statistically dependent, since some may have influenced the sequence of parameter updates that produced the others. This coupling is mediated by how strongly any individual rollout can affect subsequent iterates. At time $t_i$, a rollout generated at time $t_j < t_i$ will have contributed, on average, $(t_i - t_j) \cdot B/N$ times to the gradient updates between $t_i$ and $t_j$. As each update averages over $B$ samples, we expect the dependency to scale in $O(|t_i - t_j|/N)$. This motivates the following assumption.

**Assumption 4.3** (Variance). There exists a non-decreasing function $\sigma : \mathbb{R} \to \mathbb{R}_+$ and a coefficient $\rho \in [0, 1]$, such that for any $(t, i)$,

$$\mathbb{E}[\|\varepsilon_{t,i}\|^2] \le \sigma^2(t - t_i),$$

and for $j \ne i$,

$$\text{correlation}(\varepsilon_{t,i}, \varepsilon_{t,j}) \le \frac{\rho\, |t_i - t_j|}{N}.$$

We are now ready to state the main convergence theorem, proven in Appendix C.

**Theorem 4.4.** *Under Assumptions 4.1, 4.2 and 4.3, when the learning rate satisfies $\eta \le \min(R/(2\sqrt{2}\kappa N), L/2)$*

$$\frac{1}{T} \sum_{t=1}^{T-1} \|\nabla F(\theta_t)\|^2 \le \frac{12 F(\theta_0)}{\eta T} + 8\eta \left( \frac{4N^2 \kappa^2 \eta}{R^2} + L \right) \mathcal{V}$$

*for any $T > 1$, where $\mathcal{V}$ is a variance parameter defined as*

$$\mathcal{V} = \bar{\sigma}^2 \left( \frac{N}{R} \right) \left( \frac{1}{B} + \frac{1}{N} + \frac{\rho}{R} \right).$$

*and $\bar{\sigma}(H)$ is the average of $\sigma(1), \ldots, \sigma(H)$.*

**Theorem 4.5** (Optimal Design). *Given an asymptotically large compute budget $C$, related to the number $T$ of iterations by $C = (B + \mu R)T$, we optimize over $(\eta, N, R, B)$ the bound in Theorem 4.4. Assuming $R$ divides $N$, and relaxing integer constraints, it yields the optimal ratios*

$$N/R = x_* := \underset{x > 0}{\arg\min}\ \bar{\sigma}^2(x)(\sqrt{1/\mu} + \sqrt{\rho + 1/x})^2,$$

$$B/R = r_* := \sqrt{\mu/(\rho + 1/x_*)}.$$

*Here, $N/R$ denotes the off-policiness horizon, i.e. the maximum off-policiness of rollouts in the buffer, and $B/R$ the*

*replay ratio, i.e. the average number of times a sample is replayed over the full run.*[5]

We also provide a closed-form expression for $x_*$ in Appendix C under a power-law assumption on $\sigma$, as well as further empirical illustrations in Figure 6.

Theorem 4.5 characterizes the optimal replay-buffer design in terms of the *staleness horizon* $N/R$ and the *replay ratio* $B/R$. These ratios serve as key design levers, allowing practitioners to systematically configure the replay buffer for peak algorithmic performance. Theorem 4.5 reveals a three-way trade-off between staleness-induced noise growth ($\bar{\sigma}^2$), coupling between replayed samples and the parameter iterates ($\rho$), and the rollout-vs-training compute imbalance ($\mu$). When the compute cost of rollouts is small (small $\mu$), or when off-policy induced variance ($\bar{\sigma}$ increases fast) and correlation ($\rho$) are high, the optimal staleness horizon $x_*$ approaches zero. This suggests that in such regimes, it is more effective to remain on-policy than to utilize a replay buffer. Conversely, when rollout generation is expensive (large $\mu$) or off-policy effects are negligible ($\bar{\sigma}$ and $\rho$ are small), a replay buffer becomes optimal, characterized by a large staleness horizon and a high replay count. Overall, our theory formalizes the central trade-off studied in our experiments: replay can substantially reduce inference compute, but only up to the point where staleness-induced variance and samples-iterate correlations begin to dominate the benefit of reusing trajectories.

## 5. Experimental results

We explore how experience replay impacts accuracy and compute efficiency when training small and mid-size models with asynchronous RL fine-tuning on reasoning datasets.

### 5.1. Experimental setup

We evaluate replay buffers in the *asynchronous* setting described in Section 3, with $W$ inference workers generating rollouts and $T$ trainers performing optimization steps from a shared buffer. Unless otherwise specified, we sample from the buffer uniformly. In our primary experiments, we fine-tune Qwen3-0.6B and Qwen2.5-7B (Qwen et al., 2025) with GRPO (Shao et al., 2024) on OpenR1-Math-220k (OpenR1 et al., 2025), and evaluate on either OpenR1-Math-220k or MATH (Hendrycks et al., 2021). Unless stated explicitly, we use a learning rate of $3.37 \cdot 10^{-7}$ for Qwen3-0.6B and $6 \cdot 10^{-8}$ for Qwen2.5-7B. We plot accuracy w.r.t. either

the number of gradient steps, the compute spent (estimated with (2)) or the wall-time. All our experiments are run with at least 4 random seeds, and we report the median and the interquartile range. See Appendix E for ablations on the learning rate, and Appendix D for additional details on the setup, including the estimation of the optimal ratio $\mu$.

### 5.2. Main results

Figure 1 summarizes our central finding: *for a good choice of buffer configuration, one may save up to 40% of compute to reach a given accuracy*. For all compute budget, the accuracy achievable using experience replay is superior to that achievable with strictly on-policy training, contradicting the current paradigm. Moreover, we observe an additional benefit not predicted by our theory: using a buffer stabilizes training, preventing crashes and sometimes enabling a higher peak accuracy. These findings are confirmed on other buffer configurations, other models and other tasks in Figures 15, 16 and 17 of Appendix E.

We now run more comprehensive experiments on a smaller model to further analyze the impact of various buffer hyper-parameters and better understand these phenomena.

**Buffer Size and Off-Policiness.** The left side of Figure 3 shows the test accuracy of Qwen3-0.6B for $(W, T) = (6, 2)$ and various buffer sizes as a function of compute. We first observe that all training trajectories (with or without buffer) culminate in a global maximum accuracy, followed by a decline in performance–this is not an uncommon phenomenon in RL (see, e.g., Zheng et al., 2025).[6] We further observe that increasing the size of the buffer, hence increasing the average off-policiness of the samples, has two marked effects: it slows down the training, and it stabilizes it, leading to a potentially higher maximal accuracy that is reached later in the training run. As a secondary exploration, we trained the same model without a replay buffer and introduced various levels of off-policiness in the training distribution. The results, reported in Figure 12 in the Appendix, align with our findings and show that moderate levels of off-policiness can have a stabilizing effect on the training (independently from the use of experience replay). We hypothesize that reusing rollouts sampled from older policies regularizes the (evolving) objective function by increasing the diversity of the training distribution, and thus helps prevent overfitting. As larger models take much longer to overfit, the same effect is not visible in Figure 1.

**Replay ratio.** As the ratio $W/T$ between inference workers and trainers decreases, the compute cost of each gra-

---

[5]Note that $R/N$ is the ratio of fresh samples in the buffer, thus by contraposition $N/R$ is the number of rounds a sample will stay in the buffer. Moreover, since each sample in the buffer is associated with a sampling probability $1/N$, we sample $B$ of them in a batch, and a sample stays for $N/R$ round in the buffer, their average use over their shelf-life is $(1/N)B(N/R) = B/R$.

[6]Looking at the training accuracy (Figure 13 in Appendix E), we see that it peaks later than the test accuracy, then crashes as well, indicating that the models initially overfit before ultimately collapsing into a nonsensical policy.

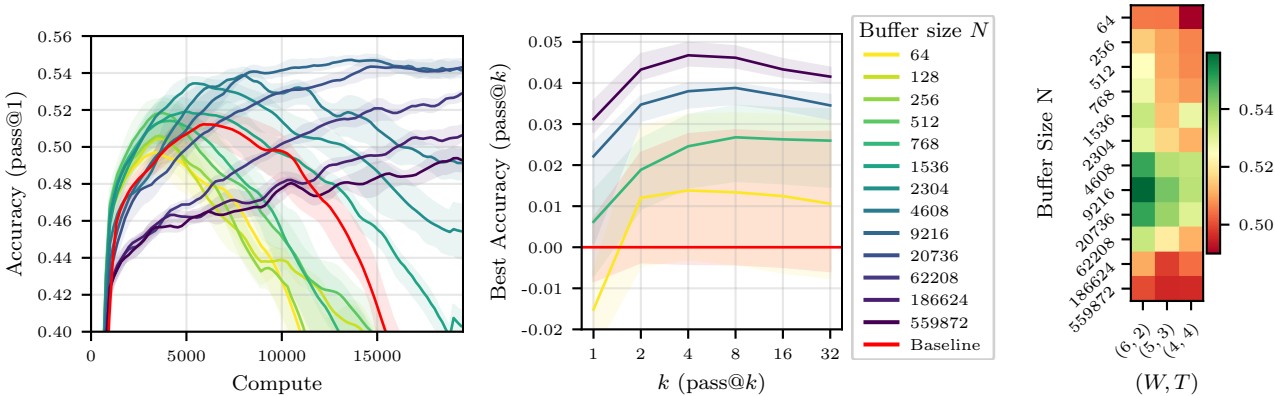

*Figure 3.* **Accuracy and Pass@**$k$ **with respect to Buffer Size.** *Left*: Test accuracy as a function of compute spent when training Qwen3-0.6B on OpenR1-Math-220k for $(W, T) = (6, 2)$ and various buffer sizes $N \in \{64, 128, 256, 512, 768, 1536, 2304, 6912, 20736\}$, as well as for a no-buffer baseline. We report the median and IQR over more than 4 seeds. Compute is normalized so that each weight update costs 1 unit for buffer configurations and 1.96 for the baseline. *Middle*: Pass@$k$ increase after training for a representative subset of these buffer configurations, relative to the baseline. *Right*: Best Accuracy achieved over entire runs for various buffer sizes and $W/T$ ratios. The compute needed to reach those accuracies is reported in Figure 14 in Appendix E.

dient update drops (Table 1), but the average replay ratio rises, going from 2.2 for $(W, T) = (6, 2)$ to 5.6 and 17.6 for $(W, T) = (5, 3)$ and $(4, 4)$ respectively. We see on the heatmap in Figure 3 that while moderate replay ratios do not adversely affect the maximal accuracy, aggressive replay eventually degrades performances (most likely due to the associated reduced *local* sample diversity, see Section 3). As shown on the more exhaustive plots for $(W, T) \in \{(5, 3), (4, 4)\}$ in Figure 13 of Appendix E, more extreme configurations can nonetheless remain attractive due to their high compute efficiency.

**Output diversity.** One can see in Figure 3 that training with experience replay can also improve the pass@k (for $k > 1$). This is true in absolute terms (i.e. the pass@k is improved), but also comparatively: using a buffer helps the pass@k for large k even more than it helps the pass@1. This shows that while the loss in diversity of the model's output distribution is a major concern in RL (Cui et al., 2025; Yue et al., 2025), experience replay can help preserve it. We attribute this phenomenon to the increased diversity of the training distribution which results from the use of older samples.

To summarize, our experiments suggest that reducing the ratio $W/T$ improves compute efficiency but worsens learning dynamics, whereas increasing the buffer size slows training while stabilizing it and helping preserve output diversity. Under suitable configurations, these effects combine to yield a net improvement across all metrics relative to strictly on-policy RL.

### 5.3. Wall-time Speed

We found compute (as defined through (2)), which isolates the algorithmic effect of experience replay (fewer rollouts per update), to be a more informative metric than wall-time, which is influenced by implementation-dependent scheduling and queuing effects. That said, we have observed that the gains in wall-speed from using a buffer in our particular setup either match or exceed the gains in compute efficiency (see Figures 10 and 11 in Appendix E).

Indeed, in asynchronous settings (described in Section 3), inference workers often stall when the transfer queue is full, and trainers stall when the queue is empty—both effects are exacerbated when reward computation introduces variable latency (as also noted by Lu et al. (2025)). This can occur even when the optimal ratio $\mu$ of trainer GPUs to inference GPUs is achieved, and is exacerbated when it is not. A replay buffer *attenuates* these stalls by decoupling production from consumption: trainers can continue optimizing even when rollout generation temporarily slows, and inference workers can continue generating rollouts even when trainers are temporarily back-pressured. This smoothing effect brought by the buffer is independent from the increase in compute efficiency discussed at length above. It can be leveraged to streamline an asynchronous RL pipeline with a ratio $W/T$ set to precisely $\mu$ while keeping the expected replay ratio equal to 1.

### 5.4. Controlling for the learning rate: optimality curves

We performed preliminary ablations (reported in Figures 8 and 9 in Appendix E) to ensure that we selected for each model the optimal learning rate for the baseline, i.e. that

which led to the highest maximum accuracy and the greatest training stability. As experience replay changes the optimization dynamics,[7] we ran further control experiments to ensure that the efficiency gains reported cannot be attributed to inadequate hyperparameters tuning. Namely, we performed an extensive sweep across learning rates and buffer configurations. For both buffer and non-buffer setups, we plot for each compute budget the best achievable accuracy (over learning rates and buffer parameters) for that budget, resulting in two *optimality curves* reported in Figure 4. We find that the best buffer configurations consistently outperform the best non-buffer configurations.

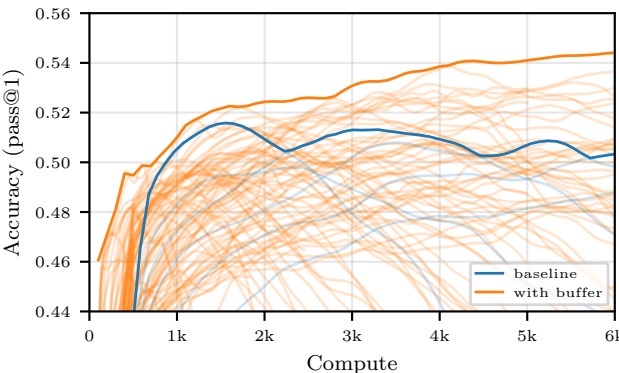

*Figure 4.* **Pareto Frontier across Hyperparameters Sweep.** Test accuracy as a function of compute spent when training Qwen3-0.6B on OpenR1-Math-220k for various learning rates ($\{1.5^i \cdot 10^{-7}\}_{i=0}^5$) and buffer configurations: no buffer (blue curves), buffer of size $\{64, 128, 256, 512, 768, 2304, 6912, 20736\}$ with $(W, T) \in \{(6, 2), (5, 3), (4, 4)\}$ (orange curves). Each curve is the median over at least 4 seeds. The two boldfaced curves delineate the Pareto frontier of each family of runs. Compute is normalized so that each weight update costs 1 unit for baseline configurations and and 0.51 for buffer configurations.

### 5.5. Further optimization: refining replay buffer design

So far, we have intentionally focused on the simplest replay buffer implementation, requiring the least deviation from the standard SOTA pipelines. We now extend our study to more exotic designs in search of further improvements, and consider two refinements. Firstly, we replace the basic sampling strategy used hitherto with a modified strategy, that we call *positive-bias sampling*: instead of keeping the freshest $N$ generated rollouts in the buffer, we keep the freshest $(1 - \delta)N$ generated rollouts along with the freshest $\delta N$ *correct* rollouts *not* included in those $(1 - \delta)N$ trajectories (an example is given in Appendix D.4), and uniformly sample from these $N$ samples. Our intuition is that the utility of correct rollouts is less affected by off-policiness.

---

[7]E.g., a (statistically unlikely) scenario where the exact same training batch is reused twice in a row would in fact be equal, up to second order terms, to a single gradient step with a learning rate twice as large.

Secondly, we replace GRPO with the AsymRE loss from Arnal et al. (2025), which has shown promises in such settings (see Appendix D). Unlike GRPO, AsymRE does not feature importance ratio correction, which is known to increase variance when off-policiness is high and does not account for subtle dependency effects when sampling from a buffer.

As showcased in Figure 5, we find that both variants lead to substantial improvements over the basic buffer implementation; larger-scale experiments are now needed to validate the robustness of these findings.

As a third refinement, we also tried sampling from a (standard) buffer uniformly *without replacement* in order to increase local diversity, but the results were inconclusive (see Figure 18).

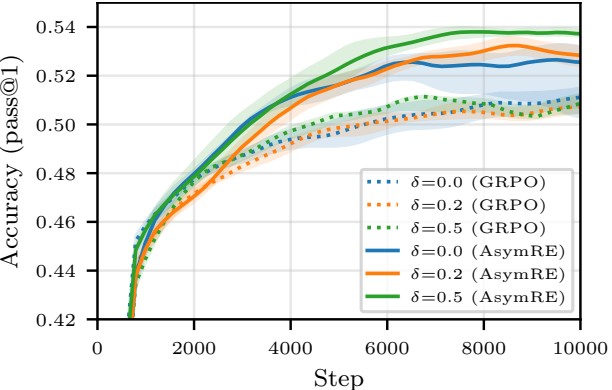

*Figure 5.* **Alternative Loss, Positive-Bias Sampling Rule.** Test accuracy as a function of training steps when training Qwen3-0.6B on OpenR1-Math-220k with a buffer of size $N = 4608$ and $(W, T) = (6, 2)$. We use either GRPO or AsymRE, and apply positive-bias sampling with coefficient $\delta \in \{0, 0.2, 0.5\}$ (note: $\delta = 0$ corresponds to standard uniform sampling).

## 6. Conclusion

In this work, we challenged the "generate-then-discard" paradigm that currently dominates LLM reinforcement learning. Through a combination of theoretical analysis and extensive empirical evaluation, we show that a well-configured replay buffer serves as a powerful lever for compute efficiency. Our theoretical framework characterizes a fundamental three-way trade-off between staleness, sample diversity, and the relative cost of inference. We show that as the computational burden of rollout generation grows, the optimal strategy shifts decisively toward experience replay. Empirically, we find that these gains are not merely theoretical: a simple replay buffer can reduce the compute budget by up to 40% while maintaining or even surpassing the accuracy of on-policy baselines. These findings suggest that maximizing performance per unit of compute, rather

than per gradient step, is a more practical objective for RL pipelines, and that experience replay is a key component in achieving this.

While our results are consistent for the model scales evaluated in this study, further work is needed to validate these efficiency gains on larger frontier models. Additionally, we believe that the Pareto frontier can be pushed further by moving beyond uniform buffers toward more sophisticated sampling rules and off-policy corrections, as well as other losses.

## Impact Statement

This paper presents work whose goal is to advance the field of Machine Learning. There are many potential societal consequences of our work, none which we feel must be specifically highlighted here.

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

# A. Extended Related Work

We provide here a more comprehensive overview of experience replay in reinforcement learning, ranging from foundational deep RL works to the most recent applications in Large Language Models.

## A.1. Experience Replay in Classical Deep RL

The concept of improving computational efficiency by storing and reusing past transitions is standard in general RL but has historically been difficult to stabilize.

- **Foundations:** Mnih et al. (2015) (DQN) demonstrated that training on samples drawn randomly from a replay buffer breaks temporal correlations in data, stabilizing the training of value functions. This became a standard component of off-policy learning.

- **Prioritized Sampling:** Schaul et al. (2015) introduced Prioritized Experience Replay (PER), improving upon uniform sampling by prioritizing transitions with high temporal-difference (TD) error, effectively focusing learning on "surprising" or difficult examples.

- **Hindsight Replay:** Andrychowicz et al. (2017) proposed Hindsight Experience Replay (HER) for goal-oriented tasks. By re-labeling failed trajectories as successful attempts towards the state they *did* reach, HER allows agents to learn from failure, significantly boosting sample efficiency in sparse-reward settings.

- **Theoretical Analysis:** Zhang & Sutton (2017) provided an early theoretical analysis of experience replay, investigating the relationship between buffer size, replay ratio, and performance, a line of inquiry we extend to the LLM setting in Section 4.

## A.2. Off-Policy Algorithms

Using a replay buffer inherently introduces off-policiness—the discrepancy between the data-generating policy and the current policy. Various algorithms have been designed to handle this:

- **Actor-Critic Methods:** DDPG (Lillicrap et al., 2015) and Soft Actor-Critic (SAC) (Haarnoja et al., 2018) are off-policy algorithms that update the policy using samples from a buffer. SAC, in particular, maximizes both expected return and entropy, stabilizing training in complex environments.

- **Off-Policy Corrections:** The Retrace algorithm (Munos et al., 2016) utilizes truncated importance sampling to safely learn from multi-step returns generated by behavioral policies. Addressing the instability of stale updates in LLMs, Zheng et al. (2025) propose second-moment constraints (M2PO) to stabilize off-policy training.

- **Recent Theoretical Advances:** More recent approaches derive consistency conditions from KL-regularized policy optimization problems (Rafailov et al., 2023; Richemond et al., 2024; Tang et al., 2025; Cohen et al., 2025), analyze the role of dataset coverage (Song et al., 2024), or propose additive renormalization of baselines (Arnal et al., 2025) to handle distribution shifts mathematically.

## A.3. Experience Replay in Modern LLM Training

While on-policy methods like PPO (Schulman et al., 2017) and GRPO (Shao et al., 2024) dominate the current LLM landscape, a wave of very recent works (2025) has begun to explore replay mechanisms. However, their goals differ significantly from ours:

- **Improving Performance via Exploration:** Bartoldson et al. (2025) use a replay buffer combined with a dedicated loss function specifically to increase exploration in sparse reward settings. Similarly, Wang et al. (2025) focus on saving successful solutions to challenging prompts ("gold samples") to facilitate reasoning breakthroughs.

- **Complex Training Pipelines:** Zhang et al. (2025) propose a two-phase training procedure where samples from an initial exploration phase are reused, while Lu et al. (2025) use a buffer to implement dynamic sampling strategies.

Unlike these works, which often introduce complex new objectives to maximize final accuracy, our work conducts a rigorous analysis of the *efficiency* trade-offs in standard pipelines with the addition of a simple replay buffer. We aim to answer *how much compute can be saved by reusing data in a standard asynchronous setup without degrading performance?*

## B. Pseudo-Code Implementation

This section provides a peudo-code implementation of the asynchronous Reinforcement Learning pipeline. This code utilizes Python's `asyncio` library to simulate the concurrent execution of inference workers ($W$) and trainers ($T$). It highlights the transition from a standard stream-based approach to the replay Buffer architecture discussed in our work.

### B.1. Queue-based Data Transfer

In baseline asynchronous RL, data typically flows through a last-in, first-out (LIFO) pipe, prioritizing the freshest samples for training. As noted in Section 3, this structure forces a tight coupling between rollout generation and consumption, where trajectories are discarded after a single update.

```python
import asyncio
import random

class QueueStructure:
    """Standard FIFO storage for on-policy rollouts."""
    def __init__(self):
        self.queue = asyncio.LifoQueue()

    async def push(self, data):
        await self.queue.put(data)

    async def sample(self, batch_size):
        # Strictly consumes data: items are removed once sampled
        return [await self.queue.get() for _ in range(batch_size)]
```

*Listing 1.* LIFO Queue implementation for on-policy streaming

### B.2. Inference Worker

The `Sampler` represents one of the $W$ inference workers. It operates in a loop, generating trajectories to be pushed into the storage structure. While the pseudo-code suggests a sequential weight update, in efficient implementations, such as the one used for this work, weights are typically updated concurrently to rollout generation to maximize throughput (i.e., the policy may change during a rollout, with later tokens generated under a different set of weights than earlier ones).

```python
class Sampler:
    def __init__(self, dump_struct):
        self.dump_struct = dump_struct

    async def run(self, dataset):
        for data in dataset:
            await self.receive_weights()
            rollout = await self.generate_rollout(data)
            await self.dump_struct.push(rollout)

        # Signal completion to the Trainer
        await self.dump_struct.push("DONE")

    async def receive_weights(self):
        """Pull latest parameters from Trainer to stay as 'on-policy' as possible."""
        ...

    async def generate_rollout(self, data):
        """Standard LLM inference step."""
        ...
```

*Listing 2.* Inference Worker (Sampler) logic

### B.3. The Consumer: Trainer

The `Trainer` represents one of the $T$ optimization units. It pulls batches of size $B$ and performs gradient updates. This loop runs concurrently with the Sampler.

```python
class Trainer:
    def __init__(self, dump_struct):
        self.dump_struct = dump_struct
        self.is_running = True

    async def run(self, batch_size):
        while self.is_running:
            batch = await self.dump_struct.sample(batch_size)

            if "DONE" in batch:
                self.is_running = False
                break

            await self.forward_backward(batch)
            await self.update_weights()

    async def forward_backward(self, batch):
        """Compute GRPO/PPO loss and gradients."""
        ...

    async def update_weights(self):
        """Apply optimizer step and broadcast new weights."""
        ...
```

*Listing 3.* Optimization Worker (Trainer) logic

### B.4. Main Orchestration

The main loop instantiates the workers. While this pseudo-code implementation uses $W = T = 1$ for simplicity, in practice more workers would operate in parallel, with a ratio of worker to trainer GPUs set to maximize GPU utilization by minimizing idle time.

```python
async def main():
    dataset = ...
    batch_size = ...
    dump_struct = ...

    sampler = Sampler(dump_struct)
    trainer = Trainer(dump_struct)

    # Launching inference and training concurrently
    await asyncio.gather(
        sampler.run(dataset),
        trainer.run(batch_size)
    )

if __name__ == "__main__":
    asyncio.run(main())
```

*Listing 4.* Asynchronous execution entry point

### B.5. The Replay Buffer: BufferStructure

A replay buffer can be implemented with minimal changes to the pipeline above. Indeed, one only needs to replace the transfer queue with a new data structure to implement experience replay. We present it below as the `BufferStructure` which will store up to $N$ buffered trajectories. Unlike the queue, this structure enables multiple samples of the same trajectory.

```
1  class BufferStructure:
2      """Experience Replay Buffer supporting random sampling."""
3      def __init__(self, buffer_size):
4          self.buffer = []
5          self.buffer_size = buffer_size
6          self.lock = asyncio.Lock()
7
8      async def push(self, data):
9          async with self.lock:
10             # FIFO eviction policy for the buffer
11             if len(self.buffer) >= self.buffer_size:
12                 self.buffer.pop(0)
13             self.buffer.append(data)
14
15     async def sample(self, batch_size):
16         async with self.lock:
17             # Sampling does not remove items from the buffer
18             return random.sample(self.buffer, batch_size)
```

*Listing 5.* Circular Replay Buffer with Random Sampling

# C. Mathematical Details

We provide additional details regarding the mathematical analysis in Section 4.

## C.1. Modeling Details

**Bias Assumption.** Assumption 4.2 can be motivated by writing the bias explicitly, using the duality bracket and any dual norms

$$\mathbb{E}[\varepsilon_{t,i} \mid \mathcal{F}_t] = \mathbb{E}_{z \sim \pi_{\theta_{t-t_i}}}[G(\theta_t, z) \mid \mathcal{F}_t] - \mathbb{E}_{z \sim \pi_{\theta_t}}[G(\theta_t, z)] = \left\langle \pi_{\theta_{t-t_i}}(\cdot \mid \mathcal{F}_t) - \pi_{\theta_t}, G(\theta_t, \cdot) \right\rangle$$
$$\leq \left\| \pi_{\theta_{t-t_i}}(\cdot \mid \mathcal{F}_t) - \pi_{\theta_t} \right\| \left\| G(\theta_t, \cdot) \right\|_*.$$

Here $\pi_{\theta_{t-t_i}}(\cdot \mid \mathcal{F}_t)$ denote the distribution of the samples under $\theta_{t-t_i}$ knowing that the future iterates up to $\theta_t$. If $z$ in position $i$ in the buffer at time $t$ was never sampled in the batches leading from $\theta_{t-t_i}$ to $\theta_t$, $\pi_{\theta_{t-t_i}}(\cdot \mid \mathcal{F}_t)$ would be equal to $\pi_{\theta_{t-t_i}}$, as knowing the iterates $\mathcal{F}_t$ would not help us reconstruct that sample. However, the more the sample was used, the more these distributions would be dissimilar. With $\kappa_0$ the average repetition of a sample in training batch from time $t_i$ to $t$, one may posit

$$\left\| \pi_{\theta_{t-t_i}}(\cdot \mid \mathcal{F}_t) - \pi_{\theta_t} \right\| \leq \kappa_0 \left\| \pi_{\theta_{t-t_i}} - \pi_{\theta_t} \right\|,$$

where $\kappa_0$ capture the measure of local diversity discussed in Section 3: the more a sample is reused on average between time $t_i$ and $t$, the bigger $\kappa$.[8] Assuming $G$ is bounded by some constant $G_\infty$, we get a bound on the bias of the form

$$\mathbb{E}[\varepsilon_{t,i} \mid \mathcal{F}_t] \leq \kappa_0 G_\infty \left\| \pi_{\theta_{t-t_i}} - \pi_{\theta_t} \right\|.$$

Finally assuming the policy is parameterized in some Lipschitz way for some constant $C$, we get

$$\mathbb{E}[\varepsilon_{t,i} \mid \mathcal{F}_t] \leq \kappa_0 G_\infty C \left\| \theta_{t-t_i} - \theta_t \right\|.$$

This motivates formally Assumption 4.2.

**Variance Assumption.** When using $z$ the $i$-th element of the buffer to estimate $\nabla F(\theta_t)$, the per-sample estimator typically includes some form of off-policy correction (explicit importance weights, clipped ratios as in PPO-style objectives, or implicit reweighting through an advantage estimator). Abstractly, one may write the estimator as $G(\theta_t, z_{t,i}) = w_{t,t_i}(z) G_0(\theta_t, z)$, where $w_{t,t_i}$ is a (possibly clipped) importance-ratio weighting between $\pi_{\theta_t}$ and $\pi_{\theta_{t_i}}$ (recall that $z$ was generated $z$ by $\pi_{\theta_{t_i}}$), and $G_0$ is a bounded-variance on-policy quantity (e.g. a score-function term times an advantage). As $\tau := t - t_i$ grows, the mismatch between $\pi_{\theta_t}$ and $\pi_{\theta_{t_i}}$ typically increases, which in turn increases the variability of importance weight $w_{t,t_i}$ and thus the variance of $G(\theta_t, z)$. This motivates an upper bound of the form $\mathbb{E}[\|\varepsilon_{t,i}\|^2] \leq \sigma^2(\tau)$ for some increasing function $\sigma^2$, which captures (in aggregate) the growth of off-policy noise with staleness.

---

[8]As such, one may want to refine $\kappa_0$ to be a function of the average number of time a sample was used between time $t_i$ and $t$, which is $(\min(t - t_i, N/R) - 1)B/N$.

**Dependencies Assumption.** In standard SGD analyses, the samples $z_t$ are i.i.d., and minibatching yields a $1/B$ variance reduction. With experience replay, however, the buffer at time $t$ is "endogenous": trajectories currently stored in the buffer may have been used in past updates, and those updates affected the parameters that later generated other trajectories that are now in the buffer. Concretely, $\varepsilon_{t,i} = G(\theta_t, z_{t,i}) - \nabla F(\theta_t)$ depends on $\theta_t$, while $\theta_t$ itself is a function of past minibatch draws; hence two buffer elements can become statistically coupled through the update trajectory that produced $\theta_t$. At a given step, a fixed element of a buffer of size $N$ is selected in expectation $B/N$ times (sampling with replacement). Over $h$ steps, it is therefore used about $hB/N$ times. Since each update is an average over $B$ samples, each occurrence contributes a factor $1/B$ to the update. Now consider two distinct buffer elements $i \neq j$ with insertion times $t_i \leq t_j$. The updates in the interval $[t_i, t_j)$ can transmit information from $z_{t,i}$ to later iterates that enter $\varepsilon_{t,j}$ (since $z_{t,j}$ is only generated at time $t_j$). Thus the strength of the coupling should generally increase with the temporal separation $|t_i - t_j|$. Aggregating algorithm-specific constants (e.g. clipping, advantage normalization, optimizer state) into a function $\rho$, this motivates

$$\text{corr}(\varepsilon_{t,i}, \varepsilon_{t,j}) \leq \frac{\rho}{N} |t_i - t_j|,$$

for $\rho$ a value in $[0, 1]$. Note that even when $t_i = t_j$, a residual dependence may remain because both trajectories can jointly influence the subsequent parameter path and hence $\theta_t$. While we omit it for simplicity, adding it would not change much the derivations.

## C.2. Proof of Convergence

Combining the $L$-smoothness Assumption 4.1 with one of Taylor expansion formulas yields

$$F(y) \leq F(x) + \langle \nabla F(x), y - x \rangle + \frac{L}{2} \|y - x\|^2.$$

Applied in $\theta_{t+1}$ and $\theta_t$

$$F(\theta_{t+1}) \leq F(\theta_t) + \langle \nabla F(\theta_t), \theta_{t+1} - \theta_t \rangle + \frac{L}{2} \|\theta_{t+1} - \theta_t\|^2.$$

With

$$\theta_{t+1} - \theta_t = -\eta g_t = -\eta(\nabla F(\theta_t) + \varepsilon_t),$$

we get

$$F(\theta_{t+1}) \leq F(\theta_t) - \eta \langle \nabla F(\theta_t), \nabla F(\theta_t) + \varepsilon_t \rangle + \frac{L\eta^2}{2} \|\nabla F(\theta_t) + \varepsilon_t\|^2.$$

Developing and rearranging leads to

$$F(\theta_{t+1}) \leq F(\theta_t) - \left( \eta - \frac{L\eta^2}{2} \right) \|\nabla F(\theta_t)\|^2 - \left( \eta - L\eta^2 \right) \langle \nabla F(\theta_t), \varepsilon_t \rangle + \frac{L\eta^2}{2} \|\varepsilon_t\|^2.$$

Summing over $t$ and rearranging with get

$$\left( \eta - \frac{L\eta^2}{2} \right) \frac{1}{T} \sum_{t=0}^{T-1} \|\nabla F(\theta_t)\|^2 \leq \frac{F(\theta_0) - F(\theta_T)}{T} - \left( \eta - L\eta^2 \right) \frac{1}{T} \sum_{t=0}^{T-1} \langle \nabla F(\theta_t), \varepsilon_t \rangle + \frac{L\eta^2}{2} \frac{1}{T} \sum_{t=0}^{T-1} \|\varepsilon_t\|^2.$$

Assuming

$$L\eta < 1/2, \tag{3}$$

we get, with $\xi$ the sign of $\sum \langle F(\theta_t), \varepsilon_t \rangle$,

$$\frac{3}{4T} \sum_{t=0}^{T-1} \|\nabla F(\theta_t)\|^2 \leq \frac{F(\theta_0) - F(\theta_T)}{\eta T} + \frac{\xi}{T} \sum_{t=0}^{T-1} \langle \nabla F(\theta_t), \varepsilon_t \rangle + \frac{L\eta}{2} \frac{1}{T} \sum_{t=0}^{T-1} \|\varepsilon_t\|^2.$$

Taking the expectation with respect to $\mathcal{F}_T$, we bound, using Cauchy-Schwarz and a Young's inequality,

$$\mathbb{E}[\langle \nabla f(\theta_t), \varepsilon_t \rangle \mid \mathcal{F}_T] = \mathbb{E}[\langle \nabla f(\theta_t), \varepsilon_t \rangle \mid \mathcal{F}_t] = \langle \nabla f(\theta_t), \mathbb{E}[\varepsilon_t \mid \mathcal{F}_t] \rangle$$

$$\leq \|\nabla f(\theta_t)\| \|\mathbb{E}[\varepsilon_t \mid \mathcal{F}_t]\| \leq \frac{1}{4} \|\nabla f(\theta_t)\|^2 + \|\mathbb{E}[\varepsilon_t \mid \mathcal{F}_t]\|^2.$$

Hence,

$$\frac{\xi}{T} \sum_{t=0}^{T-1} \langle \nabla F(\theta_t), \varepsilon_t \rangle \leq \frac{1}{4T} \sum_{t=0}^{T-1} \|\nabla f(\theta_t)\|^2 + \frac{1}{T} \sum_{t=0}^{T-1} \|\mathbb{E}[\varepsilon_t \mid \mathcal{F}_t]\|^2.$$

Plugging this into the previous inequality, we get

$$\frac{1}{2T} \sum_{t=0}^{T-1} \mathbb{E}[\|\nabla F(\theta_t)\|^2] \leq \frac{F(\theta_0) - F(\theta_T)}{\eta T} + \mathbb{E}\Big[\frac{1}{T} \sum_{t=0}^{T-1} \|\mathbb{E}[\varepsilon_t \mid \mathcal{F}_t]\|^2\Big] + \frac{L\eta}{2} \frac{1}{T} \sum_{t=0}^{T-1} \mathbb{E}[\|\varepsilon_t\|^2].$$

We need to bound the last two quantities, which we identify as the "bias", and the "variance" part.

### C.2.1. BOUND ON THE BIAS.

Under Assumption 4.2, with uniform sampling over the buffer, assuming $R$ divides $N$ for simplicity, with $H = N/R$ the staleness horizon,

$$\|\mathbb{E}[\varepsilon_t \mid \mathcal{F}_t]\|^2 = \Big\|\mathbb{E}_i[\mathbb{E}[\varepsilon_{t,i} \mid \mathcal{F}_t]]\Big\|^2 \leq \mathbb{E}_i\Big[\|\mathbb{E}[\varepsilon_{t,i} \mid \mathcal{F}_t]\|^2\Big] \leq \kappa^2 \mathbb{E}_i\big[\|\theta_t - \theta_{t_i}\|^2\big] = \frac{\kappa^2}{H} \sum_{0 \leq \tau < H} \|\theta_t - \theta_{t-\tau}\|^2.$$

The drift is controlled by the magnitude of the gradient updates,

$$\theta_t - \theta_{t-\tau} = \eta \sum_{t-\tau \leq s < t} g_s = \eta \sum_{t-\tau \leq s < t} \nabla F(\theta_s) + \eta \sum_{t-\tau \leq s < t} \varepsilon_s.$$

We proceed with the following bound

$$\|\theta_t - \theta_{t-\tau}\|^2 \leq 2\tau\eta^2 \sum_{t-\tau \leq s < t} \|\nabla F(\theta_s)\|^2 + \|\varepsilon_s\|^2 \leq 2H\eta^2 \sum_{t-H \leq s < t} \|\nabla F(\theta_s)\|^2 + \|\varepsilon_s\|^2.$$

Summing over $t$, we get

$$\frac{1}{T} \sum_{t=0}^{T-1} \|\mathbb{E}[\varepsilon_t \mid \mathcal{F}_t]\|^2 \leq 2H^2\eta^2\kappa^2 \frac{1}{T} \sum_{t=0}^{T-1} \mathbb{E}[\|\nabla F(\theta_t)\|^2 \mid \mathcal{F}_t] + \mathbb{E}[\|\varepsilon_s\|^2 \mid \mathcal{F}_t].$$

When

$$2H^2\kappa^2\eta^2 \leq 1/4, \tag{4}$$

plugging our bound on the bias into the main inequality gives

$$\frac{1}{4T} \sum_{t=0}^{T-1} \mathbb{E}[\|\nabla F(\theta_t)\|^2] \leq \frac{F(\theta_0) - F(\theta_T)}{\eta T} + \Big(\frac{2N^2\kappa^2\eta^2}{R^2} + \frac{L\eta}{2}\Big) \frac{1}{T} \sum_{t=0}^{T-1} \mathbb{E}[\|\varepsilon_t\|^2].$$

### C.2.2. REARRANGEMENT BETWEEN THE VARIANCE AND THE SECOND MOMENT

We need to bound the second-moment $\mathbb{E}[\|\varepsilon_t\|^2]$. Let introduce

$$\xi_{t,i} = \varepsilon_{t,i} - \mathbb{E}[\varepsilon_{t,i}], \qquad \xi_t = \frac{1}{B} \sum_{j \in [B]} \xi_{t,i_j}.$$

We have, reusing the previous bound on the bias, together with Eq. (4),

$$\frac{1}{T} \sum_{t=0}^{T-1} \mathbb{E}[\|\varepsilon_t\|^2] = \mathbb{E}\Big[\frac{1}{T} \sum_{t=0}^{T-1} \mathbb{E}[\|\varepsilon_t\|^2 \mid \mathcal{F}_t]\Big] = \mathbb{E}\Big[\frac{1}{T} \sum_{t=0}^{T-1} \|\mathbb{E}[\varepsilon_t \mid \mathcal{F}_t]\|^2\Big] + \frac{1}{T} \sum_{t=0}^{T-1} \mathbb{E}[\|\xi_t\|^2]$$

$$\leq \frac{1}{4T} \sum_{t=0}^{T-1} \mathbb{E}[\|\nabla F(\theta_t)\|^2 \mid \mathcal{F}_t] + \frac{1}{4T} \sum_{t=0}^{T-1} \mathbb{E}[\|\varepsilon_s\|^2 \mid \mathcal{F}_t] + \frac{1}{T} \sum_{t=0}^{T-1} \mathbb{E}[\|\xi_t\|^2]$$

Hence,

$$\frac{1}{T}\sum_{t=0}^{T-1}\mathbb{E}[\|\varepsilon_t\|^2] \le \frac{1}{3T}\sum_{t=0}^{T-1}\mathbb{E}[\|\nabla F(\theta_t)\|^2 \mid \mathcal{F}_t] + \frac{4}{3T}\sum_{t=0}^{T-1}\mathbb{E}[\|\xi_t\|^2]$$

Plugging this into the main bound, and rearranging,

$$\frac{1}{12T}\sum_{t=0}^{T-1}\mathbb{E}[\|\nabla F(\theta_t)\|^2] \le \frac{F(\theta_0) - F(\theta_T)}{\eta T} + \frac{4}{3}\left(\frac{2N^2\kappa^2\eta^2}{R^2} + \frac{L\eta}{2}\right)\frac{1}{T}\sum_{t=0}^{T-1}\mathbb{E}[\|\xi_t\|^2].$$

### C.2.3. BOUND ON THE VARIANCE

Using Assumption 4.3, we bound, with $\gamma(|t_i - t_j|)$ the correlation between $\varepsilon_{t,i}$ and $\varepsilon_{t,j}$,

$$\langle \xi_{t,i}, \xi_{t,j} \rangle] = \mathbb{P}(i = j)\mathbb{E}[\|\xi_{t,i}\|^2] + \mathbb{P}(i \ne j)\mathbb{E}[\langle \xi_{t,i}, \xi_{t,j} \rangle \mid i \ne j]$$

$$\le \mathbb{P}(i = j)\mathbb{E}[\sigma(t - t_i)^2] + \mathbb{P}(i \ne j)\mathbb{E}\Big[\gamma(t_i - t_j)\sigma(t - t_i)\sigma(t - t_j)\Big]$$

Assuming $R$ divides $N$ for simplicity, we have, with $H = N/R$,

$$\mathbb{E}[\sigma(t - t_i)^2] = \frac{1}{H}\sum_{s=0}^{H-1}\sigma(s)^2 =: \bar{\sigma}^2(H),$$

together with

$$\mathbb{E}[\gamma(|t_i - t_j|)\sigma(t - t_i)\sigma(t - t_j)] = \frac{1}{H^2}\sum_{s,s'=0}^{H-1}\gamma(|s - s'|)\sigma(s)\sigma(s') \le \frac{1}{2H^2}\sum_{s,s'=0}^{H-1}\gamma(|s - s'|)(\sigma(s)^2 + \sigma(s')^2)$$

$$= \frac{1}{H^2}\sum_{s=0}^{H-1}\sigma(s)^2\sum_{s'=0}^{H-1}\gamma(|s - s'|) \le \frac{1}{H^2}\sum_{s=0}^{H-1}\sigma(s)^2\sum_{\tau=0}^{H-1}2\gamma(\tau)$$

$$= \bar{\sigma}^2(H)\frac{1}{H}\sum_{\tau=0}^{H-1}2\gamma(\tau) =: \bar{\sigma}^2(H)\bar{\gamma}(H).$$

We deduce that

$$\mathbb{E}[\|\xi_t\|^2] = \frac{1}{B^2}\sum_{j,j'\in[B]}\mathbb{E}[\langle \xi_{t,i_j}, \xi_{i_{j'}} \rangle] = \frac{1}{B^2}\sum_{j\in[B]}\mathbb{E}[\|\xi_{t,i_j}\|^2] + \frac{1}{B^2}\sum_{j\ne j'\in[B]}\mathbb{E}[\langle \xi_{t,i_j}, \xi_{i'_j} \rangle]$$

$$= \frac{1}{B}\mathbb{E}[\|\xi_{t,i}\|^2] + \frac{(B-1)}{B}\mathbb{E}[\langle \xi_{t,i_j}, \xi_{t,i_{j'}} \rangle]$$

$$\le \frac{\bar{\sigma}^2(H)}{B} + \frac{(B-1)\bar{\sigma}^2(H)}{B}(\mathbb{P}(i_j = i_{j'}) + (1 - \mathbb{P}(i_j \ne i_{j'}))\bar{\gamma}(H))$$

In the case with replacement, we get

$$\mathbb{P}(i_j = i_{j'}) = 1/N,$$

hence

$$\mathbb{E}[\|\xi_t\|^2] \le \bar{\sigma}^2(H)\left(\frac{1}{B} + \frac{B-1}{B}\frac{1}{N} + \frac{B-1}{B}\frac{N-1}{N}\bar{\gamma}(H)\right) \le \bar{\sigma}^2\left(\frac{N}{R}\right)\left(\frac{1}{B} + \frac{1}{N} + \bar{\gamma}\left(\frac{N}{R}\right)\right).$$

With $\gamma(|t_i - t_j|) = \rho\,|t_i - t_j|\,/N$, we get

$$\bar{\gamma}(H) = \frac{2}{H}\sum_{\tau=0}^{H-1}\frac{\rho\tau}{N} = \frac{2\rho}{HN}\frac{H(H-1)}{2} = \frac{\rho(H-1)}{N} \le \frac{\rho H}{N} = \frac{\rho}{R}$$

Plugging this into the main bound, we get

$$\frac{1}{T}\sum_{t=0}^{T-1}\mathbb{E}[\|\nabla F(\theta_t)\|^2] \le 12\frac{F(\theta_0) - F(\theta_T)}{\eta T} + 8\eta\bar{\sigma}^2\left(\frac{N}{R}\right)\left(\frac{4N^2\kappa^2\eta}{R^2} + L\right)\left(\frac{1}{B} + \frac{1}{N} + \frac{\rho}{R}\right).$$

### C.3. Design Trade-Off.

#### C.3.1. SOLUTION WITH SPECIFYING $\sigma$

Using that the number of gradient steps $T$ is a direct function of the compute $C = cT$, where $c = (B + \mu R)$, aiming to minimize the right hand-side in convergence bound of Theorem 4.4 provides design consideration for the buffer parameter $R$, $B$, $N$. As the compute goes to infinity, we notice that the optimal learning rate (solving a third degree polynomial equation) goes to zero. As such, the term in $\kappa^2 \eta$ becomes negligible in front of $L$ asymptotically. In this case, our analysis suggests to design of the buffer by optimizing for $R$, $N$ and $B$ in order to minimize

$$\mathcal{J}_0(R, B, N, \eta; \mu, \bar{\sigma}, \rho, \kappa, C) = \frac{12F(\theta_0)(B + \mu R)}{C\eta} + 8L\eta\bar{\sigma}^2\left(\frac{N}{R}\right)\left(\frac{1}{B} + \frac{1}{N} + \frac{\rho}{R}\right)$$

Optimizing in $\eta$ gives

$$\mathcal{J}_0(R, B, N, \eta_*; \mu, \bar{\sigma}, \rho, \kappa, C) = \frac{4\sqrt{6}\sqrt{LF(\theta_0)}}{\sqrt{C}}\sqrt{(B + \mu R)\bar{\sigma}^2\left(\frac{N}{R}\right)\left(\frac{1}{B} + \frac{1}{N} + \frac{\rho}{R}\right)}.$$

Hence, we can simplify our minimization goal by aiming to minimize

$$\mathcal{J}(R, B, N; \mu, \bar{\sigma}, \rho) = (B + \mu R)\bar{\sigma}^2\left(\frac{N}{R}\right)\left(\frac{1}{B} + \frac{1}{N} + \frac{\rho}{R}\right)$$

Let us introduce the staleness horizon $x = N/R$, which corresponds to the maximum staleness of trajectories in the buffer.

$$\mathcal{J} = \bar{\sigma}^2(x)(B + \mu R)\left(\frac{1}{B} + \frac{1}{xR} + \frac{\rho}{R}\right)$$

Let us introduce the replay ratio $y$, which is the average number of time a sample will be replayed during the SGD trajectory. Since a sample $z$ stays for $x$ iteration in the buffer, that at each iteration $B$ samples are extracted from the buffer, and that the sampling are independent between steps, the replay ratio is expressed as

$$y = \frac{N}{R} \times \mathbb{E}\left[\sum_{i \in [B]} \mathbb{I}\{z_{t,i} = z\}\right] = \frac{N}{R}\frac{B}{N} = \frac{B}{R}.$$

Using this ratio, we get

$$\mathcal{J} = \bar{\sigma}^2(x)(1 + y/\mu)\left(\frac{1}{y} + \frac{1}{x} + \rho\right)$$

We aim to minimize $\mathcal{J}(x, y)$ over the domain $\mathcal{D} = (0, +\infty)^2$. Since $\mathcal{J}$ is continuous, and tends to infinity on the border of the domain $\mathcal{D}$, it achieves its minimum for some $(x_*, y_*) \in \mathcal{D}$ (not necessarily unique). Moreover, since $\mathcal{J}$ is infinitely differentiable on its domain, $y_*$ is characterized by $\partial_y \mathcal{J}(y, x_*) = 0$, which leads to

$$0 = \left(\frac{1}{y_*} + \frac{1}{x_*} + \rho\right)/\mu - (1 + y_*/\mu)\frac{1}{y_*^2} = \left(\frac{1}{x_*} + \rho\right)/\mu - \frac{1}{y_*^2}.$$

Hence,

$$y_* = \frac{\sqrt{\mu}}{\sqrt{\left(\rho + \frac{1}{x_*}\right)}}.$$

Plugging this expression back into $\mathcal{J}$ gives a one-dimensional objective in $x$,

$$\mathcal{I}(x) := \mathcal{J}(x, \sqrt{\mu}/\sqrt{(\rho + 1/x)}) = \bar{\sigma}^2(x)\left(1 + \frac{1}{\sqrt{\mu\left(\rho + \frac{1}{x}\right)}}\right)\left(\sqrt{\left(\rho + \frac{1}{x}\right)}/\mu + \frac{1}{x} + \rho\right)$$

$$= \bar{\sigma}^2(x)\left(\frac{1}{\sqrt{\mu}} + \sqrt{\rho + \frac{1}{x}}\right)^2.$$

where the last equality follows from the fact that for $a = (\rho + 1/x)$,

$$\left(1 + \frac{1}{\sqrt{\mu a}}\right)\left(\sqrt{\frac{a}{\mu}} + a\right) = \sqrt{\frac{a}{\mu}} + a + \frac{1}{\mu} + \sqrt{\frac{a}{\mu}} = \left(\frac{1}{\mu} + \sqrt{a}\right)^2.$$

Hence the remaining design choice is

$$x_* \in \arg\min_x \ \bar{\sigma}^2(x)\left(\frac{1}{\sqrt{\mu}} + \sqrt{\rho + \frac{1}{x}}\right)^2.$$

Note that we omitted integers and divisibility constraints, which would leads to a constrained version of the solution provided above.

**Remark on "Under Specifications"** Note our analysis reduces the parameterization of $\mathcal{J}$ from three variables $(B, N, T)$ to only two ratios $(x, y)$. This reduction follows from the homogeneity of $\mathcal{J}$ under the scaling transformation $(B, N, R) \mapsto (\alpha B, \alpha N, \alpha R)$ for any $\alpha > 0$. As a consequence, Theorem 4.5 characterizes optimal ratios rather than prescribing absolute values (e.g., a specific batch size), which may at first appear puzzling.

However, the scale invariance only holds in the asymptotic regime. Entering this regime requires the number of gradient steps $T = C/(B + \mu R)$ to be sufficiently large, thus imposing an upper bound on $B + \mu R$ for a fixed compute budget $C$. Similarly, integer constraints and divisibility assumptions imposes lower bounds on $B$, $N$, and $R$. In practice, precise finite-time bound would introduce additional quantities, which would break the homogeneity, and provide clear indications on the optimal batch size.

### C.3.2. CLOSED-FORM SOLUTION WITH POWER-LAW VARIANCE

Let us specify the variance profile as a power law:

$$\sigma(x) = \left(\frac{x}{\tau}\right)^\alpha$$

for some coefficients $\tau$ and $\alpha$. Using the integral approximation $\sum_{s=0}^{H-1} s^p \approx \frac{H^{p+1}}{p+1}$, we compute the average variance $\bar{\sigma}^2(H)$:

$$\bar{\sigma}^2(H) = \frac{1}{H}\sum_{s=0}^{H-1}\left(\frac{s}{\tau}\right)^{2\alpha} \approx \frac{1}{H\tau^{2\alpha}}\frac{H^{2\alpha+1}}{2\alpha+1} = \frac{1}{2\alpha+1}\left(\frac{H}{\tau}\right)^{2\alpha}.$$

Recall that $x = N/R = H$. Substituting this into the design objective $\mathcal{I}(x)$, we aim to minimize

$$\mathcal{I}(x) = \frac{x^{2\alpha}}{(2\alpha+1)\tau^{2\alpha}}\left(\frac{1}{\sqrt{\mu}} + \sqrt{\rho + \frac{1}{x}}\right)^2.$$

Dropping constant multiplicative factors, this is equivalent to minimizing the simplified function

$$\mathcal{K}(x) = x^\alpha \left(\frac{1}{\sqrt{\mu}} + \sqrt{\rho + \frac{1}{x}}\right). \tag{5}$$

The first-order optimality condition $\mathcal{K}'(x) = 0$ yields

$$\alpha x^{\alpha-1}\left(\frac{1}{\sqrt{\mu}} + \sqrt{\rho + \frac{1}{x}}\right) + x^\alpha \frac{1}{2\sqrt{\rho + \frac{1}{x}}}\left(-\frac{1}{x^2}\right) = 0.$$

Multiplying by $2x^{2-\alpha}\sqrt{\rho + 1/x}$, we obtain the algebraic equation

$$2\alpha x\left(\frac{1}{\sqrt{\mu}}\sqrt{\rho + \frac{1}{x}} + \rho + \frac{1}{x}\right) = 1 \iff 2\alpha\sqrt{\frac{\rho x^2 + x}{\mu}} = 1 - 2\alpha - 2\alpha\rho x.$$

Squaring both sides leads to a quadratic equation $Ax^2 + Bx + C = 0$ governing the optimal staleness $x_*$:

$$4\alpha^2(\rho x^2 + x)/\mu = (1 - 2\alpha - 2\alpha\rho x)^2$$
$$\iff \underbrace{4\alpha^2\rho(1/\mu - \rho)}_{A} x^2 + \underbrace{4\alpha(\alpha/\mu + \rho(1 - 2\alpha))}_{B} x \underbrace{-(1 - 2\alpha)^2}_{C} = 0.$$

Solving for the positive root, and noting that the discriminant $\Delta = B^2 - 4AC$ simplifies to $\Delta = 16\alpha^2\mu(\alpha^2/\mu + \rho(1 - 2\alpha))$, the optimal staleness horizon is given explicitly by

$$x_* = \frac{-(\alpha/\mu + \rho(1 - 2\alpha)) + \sqrt{\alpha^2/\mu^2 + \rho(1 - 2\alpha)/\mu}}{2\alpha\rho(1/\mu - \rho)}.$$

To find the optimal replay ratio $y_*$, we avoid substituting the complex closed-form of $x_*$ and instead exploit the optimality conditions directly. Recall the relationship characterizing $y_*$:

$$y_* = \frac{1}{\sqrt{(\rho + 1/x_*)/\mu}} \implies \rho + \frac{1}{x_*} = \frac{\mu}{y_*^2}.$$

This allows us to express the staleness $x_*$ strictly as a function of $y_*$:

$$\frac{1}{x_*} = \frac{\mu}{y_*^2} - \rho = \frac{\mu - \rho y_*^2}{y_*^2} \implies x_* = \frac{y_*^2}{\mu - \rho y_*^2}.$$

Substituting the term $\sqrt{\rho + 1/x_*} = \frac{\sqrt{\mu}}{y_*}$ into the first-order optimality condition derived for $x_*$:

$$1 = 2\alpha x_* \left( \frac{1}{\sqrt{\mu}} \sqrt{\rho + \frac{1}{x_*}} + \rho + \frac{1}{x_*} \right) = 2\alpha x_* \left( \frac{1}{y_*} + \frac{\mu}{y_*^2} \right).$$

Simplifying the term in the parenthesis yields $= 1$, or equivalently:

$$x_* = \frac{y_*^2}{2\alpha(\mu + y_*)}.$$

Equating the two characterization of $x_*$ gives

$$\frac{y_*^2}{\mu - \rho y_*^2} = \frac{y_*^2}{2\alpha(\mu + y_*)} \iff \mu - \rho y_*^2 = 2\alpha(\mu + y_*).$$

Rearranging terms yields a quadratic equation in $y_*$:

$$\rho y_*^2 + 2\alpha y_* + \mu(2\alpha - 1) = 0.$$

Assuming $\alpha < 1/2$, the constant term $\mu(2\alpha - 1)$ is negative, guaranteeing a unique positive solution:

$$y_* = \frac{-\alpha + \sqrt{\alpha^2 + \mu\rho(1 - 2\alpha)}}{\rho}.$$

An illustration of the formula for $x_*$ and $y_*$ is provided in Figure 6, and the function $x \mapsto \mathcal{K}(x)$ (as well as the optimum value $x_*$) is shown in Figure 7.

## D. Experimental details

We provide additional details regarding our experimental setup.

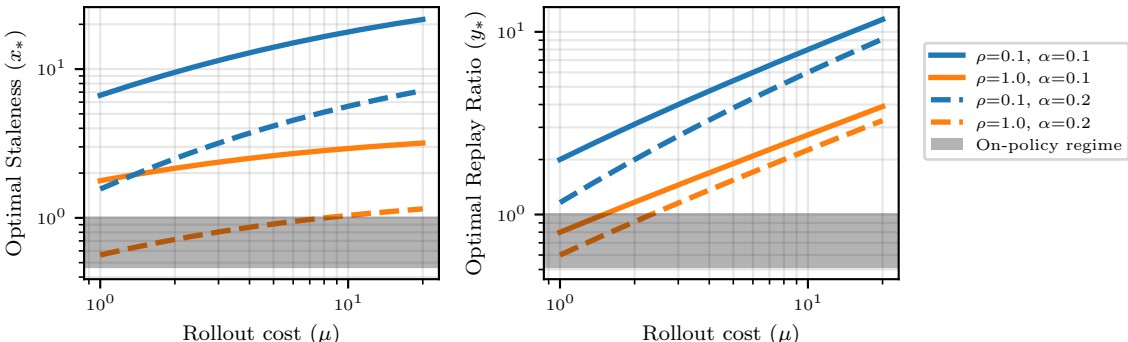

Figure 6. **Optimal Staleness and Replay Ratio as a function of Rollout Cost ($\mu$).** As $\mu$ increase, we see that it is better to increase the staleness horizon $x_* = N/R$, and the replay ratio ($y_* = B/R$). This also the case when the variance $\alpha$ or the correlation $\rho$ decreases.

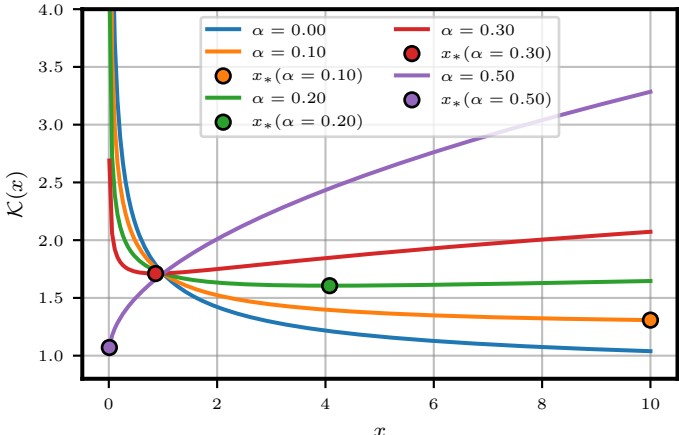

Figure 7. **Function** $x \mapsto \mathcal{K}(x)$ (which is defined in Eq. (5) and corresponds to $\mathcal{J}$ for the specific choice $\sigma(x) = (x/\tau)^\alpha$) as a function of the staleness horizon $x = N/R$, for different values of $\alpha \in [0, 1/2]$, and the corresponding optimal values of $x_*$.

### D.1. Hardware and parallelism

We use Nvidia H100 and H200 GPUs.

Our experiments are run on either $1, 2$ or $4$ 8-GPUs nodes, with data parallelism and without tensor parallelism. When describing a buffer experiment ran on more than $1$ node, we report $T$ and $W$ divided by the number of nodes; in other words, we describe an experiment run on 16 GPUs with $4$ trainer GPUs and $12$ inference GPUs as $(6, 2)$ rather than $(12, 4)$. We do so to simplify notations, and because increasing the number of nodes while keeping the same ratio $W/T$ does not impact any of the relevant quantities (size of the buffer, replay ratio, off-policiness): the training dynamics remain the same (up to essentially random effects linked to inter-nodes communications), and the training is accelerated with respect to wall-time, which we do not take into account when estimating compute (in other words, we consider that the cost of a gradient step is not affected by the number of nodes).

Our non-buffer experiments are run with $(W, T) \in \{(4, 4), (5, 3), (6, 2)\}$: though we find that the theoretical optimal ratio $\mu$ is closer to $W/T = 5$, ratios closer to $1$ are in practice better when training on a small number of GPUs (e.g. $8$ or $16$). This is because letting $T$ be very small (e.g. $T \in \{1, 2\}$) forces the maximum micro-batch size to also be very small, while large micro-batch sizes are needed to leverage parallelism-based optimizations.

### D.2. Optimization and general hyperparameters

We train using the Adam (Kingma & Ba, 2014) optimizer with constant learning rates. We use a batch size of $60$, except in the few runs for which $T = 7$, for which we let the batch size be $63$ (as it must be divisible by the number of trainer GPUs).

Unless otherwise specified, we use a learning rate of $6.8 \cdot 10^{-8}$ for Qwen2.5-7B and of $3.37 \cdot 10^{-7}$ for Qwen3-0.6B.

We use the following GRPO implementation (see (Shao et al., 2024)):

$$\mathcal{J}_{GRPO}(\theta) = \mathbb{E}_{q \sim \mathcal{D}, z}\Big[ \min\Big( \frac{\pi_\theta(z|q)}{\pi_{\theta_{old}}(z|q)} A, \mathrm{clip}\Big( \frac{\pi_\theta(z|q)}{\pi_{\theta_{old}}(z|q)}, 1 - \varepsilon_{\text{low}}, 1 + \varepsilon_{\text{high}} \Big) A \Big) \Big],$$

where $q$ is a prompt sampled from a training distribution $\mathcal{D}$ and $z$ is a rollout sampled from the buffer following the chosen sampling strategy. Both the probability $\pi_{\theta_{old}}(z|q)$ and the advantage $A$ of $z$ are computed at the time when $z$ is first generated. More specifically, a group of $G$ rollouts $z_1, \ldots, z_G$ is generated by the inference workers for each prompt $q$, and the advantage $A_i$ of $z_i$ is defined as

$$A_i = \frac{r_i - \text{mean}(\{r(z_1, q), r(z_2, q), \cdots, r(z_G, q)\})}{\text{std}(\{r(z_1, q), r(z_2, q), \cdots, r(z_G, q)\})}. \tag{6}$$

In other words, the advantage is computed when the rollout is generated (and not when it is used to compose a gradient update).

In particular, we do not include a KL regularization term, as recent research suggests that it does not improve performance (see e.g. Yu et al. (2025)). We let $\varepsilon_{\text{low}} = \varepsilon_{\text{high}} = 0.2$, and we let the group size $G$ be equal to 16. Note that when this loss is combined with a buffer, it can be shown that the joint distribution over the current training batch (which is assembled by sampling from the replay buffer) is *not* corrected in expectation by the importance sampling factor $\frac{\pi_\theta(z|q)}{\pi_{\theta_{old}}(z|q)}$ (even without taking the clipping into account).

We also consider the AsymRE objective function from Arnal et al. (2025), expressed using the same notations as

$$J_{AsymRE}(\theta) = \mathbb{E}_{q \sim \mathcal{D}, z}\Big[ \frac{1}{G} \sum_{i=1}^{G} (r(z, q) - (\hat{V} + \delta V)) \log(\pi_\theta(z|q)) \Big],$$

where $\hat{V} := \text{mean}(\{r(z_1, q), r(z_2, q), \cdots, r(z_G, q)\})$ if $z_1, \ldots, z_G$ is the group of generated rollouts to which $z$ belongs (see above) and $\delta V = -0.1$.

We train Qwen3-0.6B without weight tying.

We use a temperature of 1 when generating training samples, of 0.1 when evaluating pass@1 (with top_p = 0.95), and of 1 when evaluating pass@k with $k > 1$ (with top_p = 0.95).

### D.3. Metrics

**Compute** Our abstract measure of compute is in closest correspondence to the notion of FLOPS, but we make throughout the text the following implicit assumptions, which are never completely realized in practice:

- We are in an optimized settings in which there is a direct correspondence between FLOPS and GPU work time, except when a GPU is idle because it is waiting on the work of other GPUs,

- Tasks can be continuously parallelized; in other words, there are no boundaries effects due to the discrete nature of the number of samples and GPUs, and

- When parallelizing a task between $K$ GPUs, the total compute spent is not a function of $K$.

In particular, we ignore the effects of important implementation details, such as tensor parallelism, data parallelism, sharding, etc.

*Steps-since-last-use* We define in greater detail the steps-since-last-use metric reported in Figure 2. In the context of this paragraph, we use the term "rollout" to refer to a given data point, and the term "sample" to refer to a data point *as it appears in a gradient descent batch*. Each rollout (a given sequence of tokens) can correspond to zero, one or several samples belonging to one or several batches depending on how often it was sampled from the buffer.

- We order all samples used during a training trajectory:

- For every batch $B$, we pick a random ordering of the samples of $B$.
- If batch $B$ was processed before batch $B'$, then $z < z'$ for any $z \in B, z' \in B'$.

- Whenever a rollout appears as a sample for the first time according to this global order, we associate the value "new" to the sample.

- If a sample $z$ corresponds to a rollout that has already given rise to an earlier sample $z'$, then we associate to $z$ the number of gradient steps taken since $z'$.

As an illustration, let us assume that a rollout gives rise to exactly four samples: $z_1 \in B_3$ and $z_2, z_3, z_4 \in B_5$, where the numbering of the samples reflect their ordering and the batch $B_i$ was used at time $i$. Then $z_1$ is mapped to "new", $z_2$ to 2, $z_3$ to 0 and $z_4$ to 0. In Figure 2, we plot the histogram of the values taken by steps-since-last over all samples of each trajectory considered.

*Pass@k* The pass@k curve from Figure 3 is computed as follows: for a given $k$ and a given choice of hyperparameters (with or without buffer, etc.), the median over the random seeds of the pass@k training curve is computed. We then report the maximum of this median curve over the training trajectory, as well as the IQR at the step where the maximum is reached. In particular, the corresponding training step is in general not the same for distinct choices of $k$.

### D.4. Buffer-specific aspects

*Compute ratio $\gamma$*

To estimate the compute cost of a parameter update in a buffer configuration with $T$ trainer GPUs and $W$ inference GPUs, we use the compute ratio

$$\gamma = \frac{1 + W/T}{1 + \mu},$$

defined in Equation (2). This quantity depends in turn on the optimal ratio $\mu$, defined as the compute cost of generating a rollout divided by the compute cost of processing it through a gradient update; equivalently, $\mu$ is the exact number of inference GPUs for each trainer GPUs required so that there is no downtime.

This quantity depends on the model, dataset, implementation details and hardware, as well as on some parameter choices (such as the batch size). Consider a training run using a replay buffer, and the following quantities:

- $K_{\text{training}}$ the number of non-unique rollouts processed through backpropagation over the entire run, i.e. the number of gradient steps multiplied by the batch size,

- $K_{\text{inference}}$ the number of unique rollouts generated by the inference GPUs over the entire run,

- $T$ the number of trainer GPUs, and

- $W$ the number of inference GPUs.

Each trainer GPU will have processed $K_{\text{training}}/T$ rollouts, and each inference GPU will have generated $K_{\text{inference}}/W$ rollouts on average. As inference and trainer GPUs work independently from each other when using a replay buffer and do not suffer any downtime, we can consider that this is a fair measure of their relative speed (or equivalently of the relative compute cost of training vs inference), and use it to estimate $\mu$:

$$\mu \cong \frac{K_{\text{training}}/T}{K_{\text{inference}}/W}.$$

To make this estimate more precise, we use the median value over several random seeds.

We report in the table below our estimates of the coefficients $\mu$ for the various models featured in our experiments.

We provide in Table 3 the $\gamma$ values corresponding to Qwen3-0.6B.

*Sharded buffers* In our concrete implementation, each trainer GPU maintains its own separate replay buffer. In other words, newly generated rollouts are added to the replay buffers of the various trainer GPUs (each a list of trajectories) in a balanced

| **Model** | Median $\mu$ | IQR | # Independent runs |
|---|---|---|---|
| Qwen3-0.6B | 6.84 | $[6.45, 7.07]$ | 242 |
| Qwen2.5-7B | 5.28 | $[5.12, 5.48]$ | 129 |

*Table 2.* Estimates of $\mu$ for various models, computed as the median over several independent runs. We also provide the interquartile range over those runs.

| **(W, T)** | (7,1) | (6,2) | (5,3) | (4,4) | (2,6) | (1,7) |
|---|---|---|---|---|---|---|
| $\gamma$ | 1.02 | 0.41 | 0.34 | 0.26 | 0.17 | 0.15 |

*Table 3.* $\gamma$ for various values of $(W, T)$ and an estimated $\mu = 6.84$ for Qwen3-0.6B.

way, with each rollout being added to the replay buffer of a single trainer GPU. Each trainer GPU, when creating a sub-batch from which it will compute a gradient (which will then be averaged with the gradients of other trainer GPUs), samples only from its own buffer. We always report the total buffer size $N$, which is the sum over the $T$ trainer GPUs of the sizes $N/T$ of their separate buffers. Our preliminary experiments suggest that this design choice has little impact.

*Positive-bias sampling* We introduced in Section 5 an alternative buffer strategy, which we call positive-bias sampling: instead of keeping the freshest $N$ generated rollouts in the buffer, we keep the freshest $(1 - \delta)N$ generated rollouts along with the freshest $\delta N$ correct rollouts not included in those $(1 - \delta)N$ trajectories. As an example, if $N = 8$, $\delta = 0.75$ and the last rollouts to be produced are

$$\ldots z_{t-9} \; z_{t-8} \; z_{t-7} \; z_{t-6} \; z_{t-5} \; z_{t-4} \; z_{t-3} \; z_{t-2} \; z_{t-1} \; z_t,$$

where incorrect and correct rollouts are shown in red and green respectively, then the buffer at time $t$ is equal to

$$z_{t-8} \; z_{t-7} \; z_{t-5} \; z_{t-4} \; z_{t-3} \; z_{t-2} \; z_{t-1} \; z_t$$

# E. Additional Experimental Results

We provide various additional experimental results:

- In Figures 8 and 9, we run ablations to select the best learning rate for Qwen3-0.6B and Qwen2.5-7B. We see that $3.37 \cdot 10^{-7}$, respectively $6.810^{-8}$, achieve the best balance between speed and stability.

- We report accuracy with respect to wall-time for Qwen3-0.6B and Qwen2.5-7B in Figures 10 and 11.

- We study the impact of off-policiness in no-buffer configurations in Figure 12.

- We report accuracy with respect to compute and training dynamics for Qwen3-0.6B with various buffer sizes and $W/T$ ratios in 13. We also report the best accuracies achieved and the corresponding compute costs in Figure 14.

- We test our methods on other models and tasks in Figures16 (Qwen3-8B on Lean coding tasks) and 17 (Llama 3.2 3B on OpenR1-Math-220k).

- In complement to Figure 1, we report in Figure 15 the results for additional buffer configurations for Qwen2.5-7B.

- We show that alternative sampling strategies based on sampling without replacement do not have a clear impact in Figure 18.

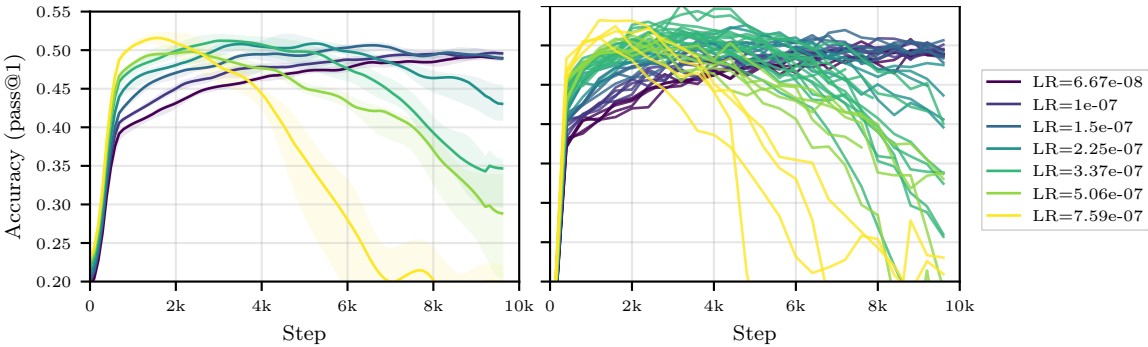

*Figure 8.* **Learning Rate Ablations for Qwen3-0.6B.** Test accuracy as a function of the number of steps when training Qwen3-0.6B on OpenR1-Math-220k with various learning rates LR with at least 4 seeds per configuration. We show the median and IQR over the seeds on the left, and all seeds separately on the right.

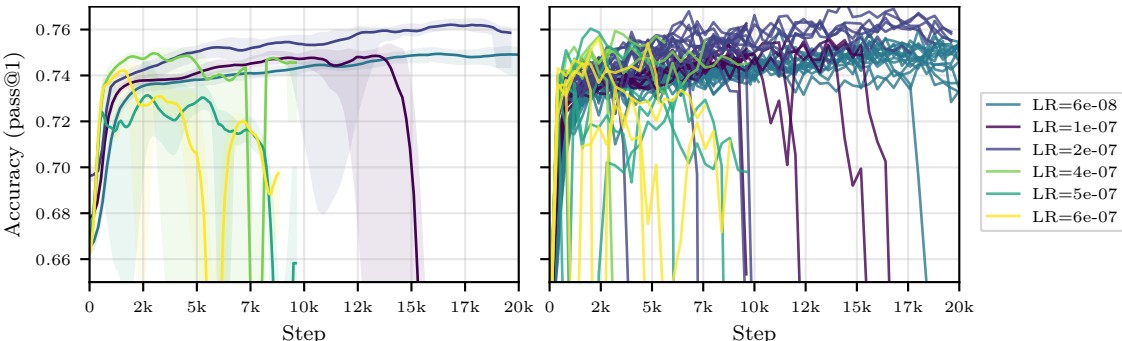

*Figure 9.* **Learning Rate Ablations for Qwen2.5-7B.** Test accuracy on MATH as a function of the number of steps when training Qwen2.5-7B on OpenR1-Math-220k with various learning rates LR with at least 4 seeds per configuration. We show the median and IQR over the seeds on the left, and all seeds separately on the right. Note the frequent crashes when $LR > 6.8 \cdot 10^{-8}$.

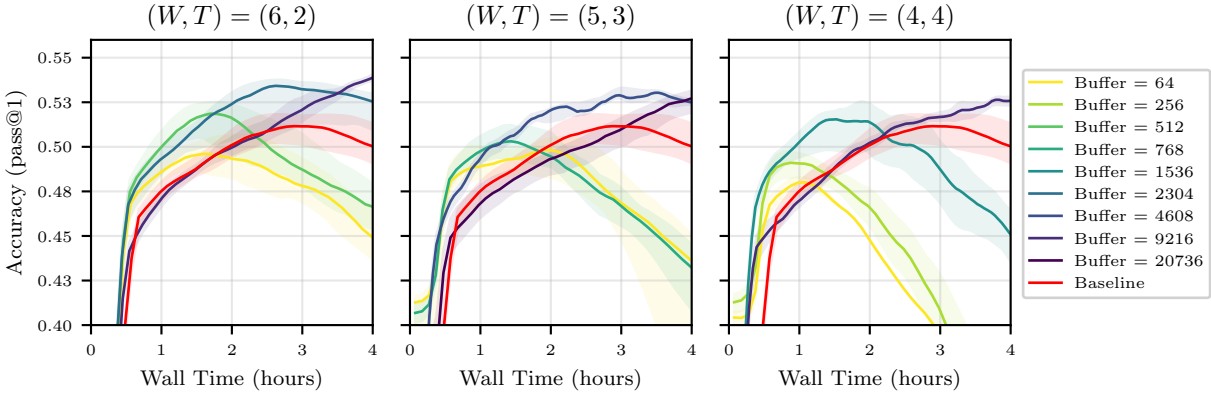

*Figure 10.* **Wall-time efficiency for Qwen3-0.6B** Test accuracy as a function of wall-time when training Qwen3-0.6B on OpenR1-Math-220k for the no-buffer baseline (orange curve) and various buffer configurations. We report the median and the IQR over at least 4 seeds per curve.

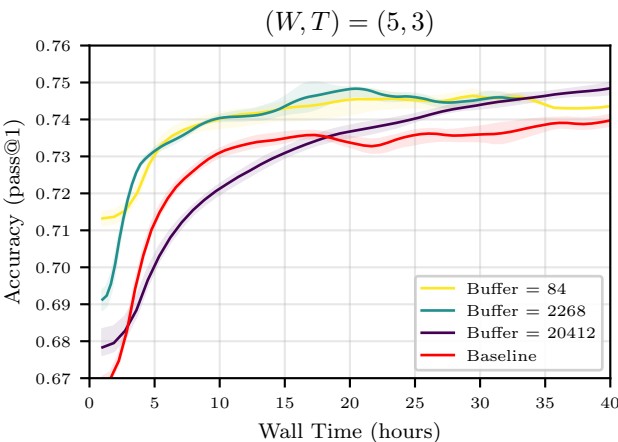

*Figure 11.* **Wall-time efficiency for Qwen2.5-7B** Test accuracy on MATH as a function of wall-time when training Qwen2.5-7B on OpenR1-Math-220k for the no-buffer baseline (orange curve) and various buffer configurations. We report the median and the IQR over at least 4 seeds per curve.

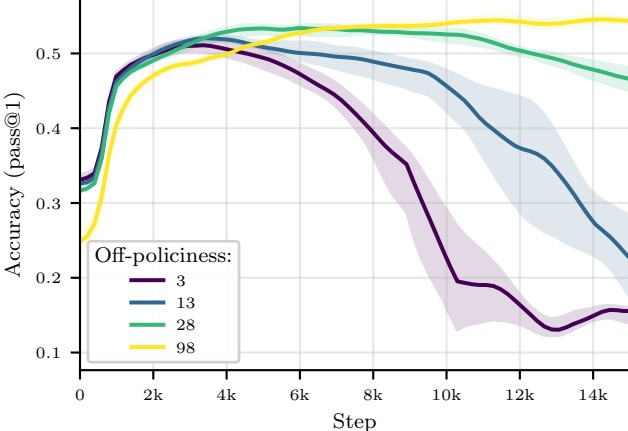

*Figure 12.* **Impact of off-policiness.** We train Qwen3-0.6B on OpenR1-Math-220k without a buffer and we artificially introduce various levels of off-policiness by reducing the frequency at which the model's weights used by the inference workers to generate rollouts are updated. We label each curve with the median level of off-policiness over all rollouts used, and plot the median test accuracy and its IQR as a function of the number of training steps over at least 4 seeds per curve.

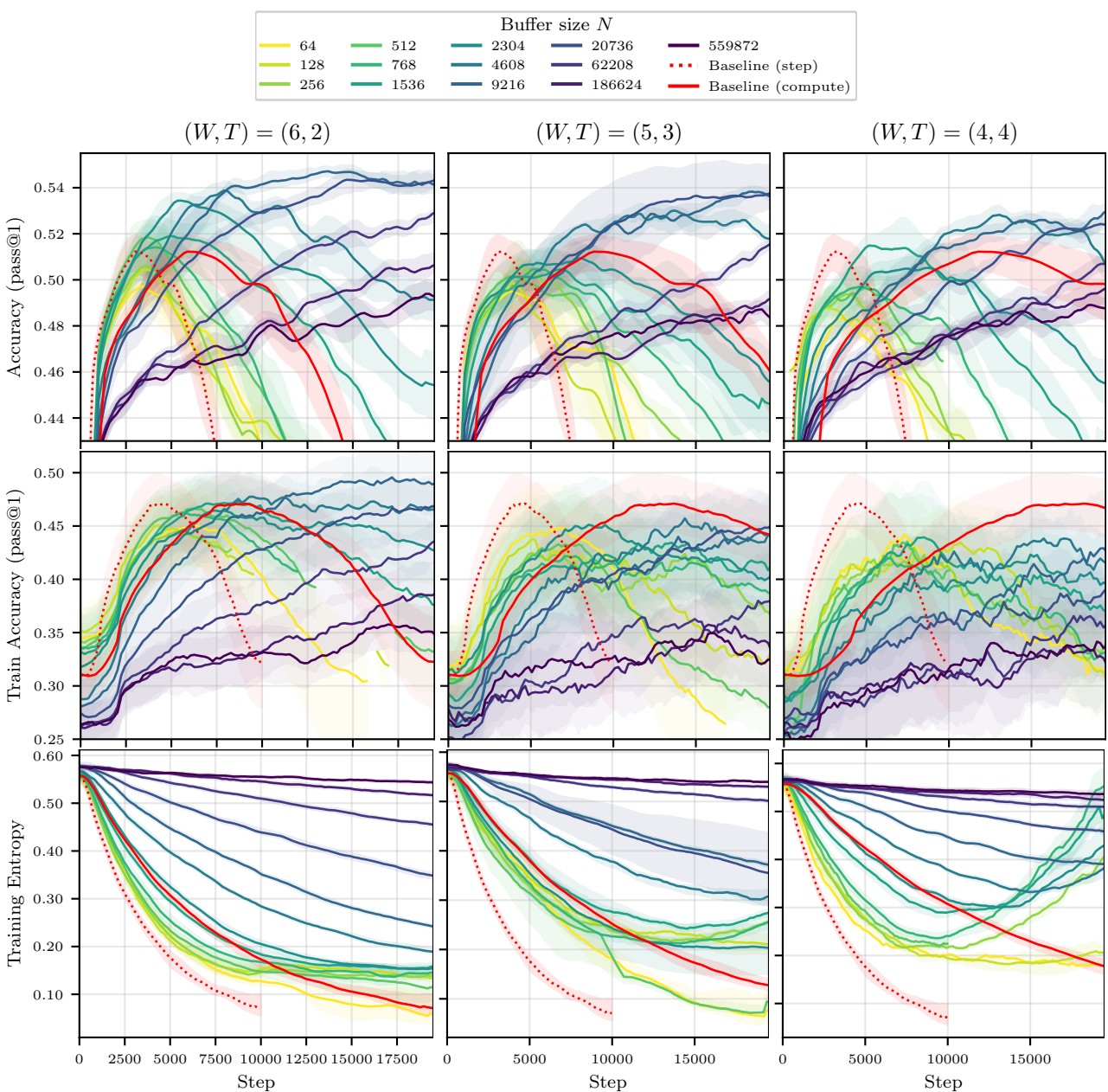

*Figure 13.* **Test, Train and Entropy Dynamics.** We train Qwen3-0.6B on OpenR1-Math-220k with a buffer for $(W, T) \in \{(6, 2), (5, 3), (4, 4)\}$ and various buffer sizes. We report the test accuracy (top), the training accuracy (middle, smoothed using a sliding window), and the training entropy (bottom) as a function of the number of training steps. Note that the training entropy is computed over the batches used by the trainers to compute gradient updates; as using a buffer implies reusing samples generated by outdated policies, it is expected that the reported entropy would be much higher. We also report two baseline curves, corresponding to non-buffer configurations: one is plotted with respect to the number of steps, while the other is rescaled to be at compute-parity with the buffer configurations (i.e. so that an x-axis unit represents the same amount of compute). Each curve is the median (along with its IQR) over at least 4 seeds.

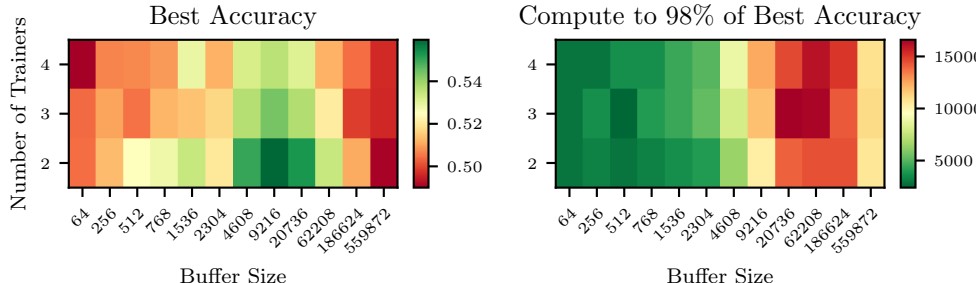

*Figure 14.* **Accuracy and Speed with respect to Design Choices.** We train Qwen3-0.6B on OpenR1-Math-220k for $(W, T) \in \{(6, 2), (5, 3), (4, 4)\}$ and various buffer sizes. We report on the right the median best test accuracy achieved over each training run (in other words, the median over the seeds of the best accuracy achieved for each seed), and on the left the median amount of compute that was needed to first reach 98% of that score.

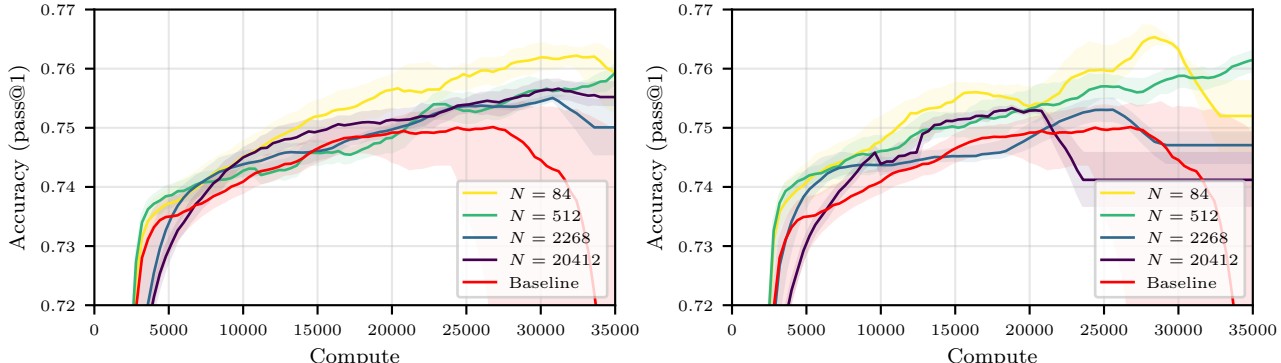

*Figure 15.* **Additional results for Qwen2.5-7B.** Accuracy on MATH as a function of compute spent when training Qwen2.5-7B on OpenR1-Math-220k for the no-buffer baseline (orange curve) and a buffer of size $N \in \{84, 512, 2268, 20412\}$ with $(W, T)$ equal to $(6, 2)$ (left) or $(5, 3)$ (right). We report the median and IQR over more than 4 seeds. Compute is calibrated so that a single weight update for the baseline costs 1 unit.

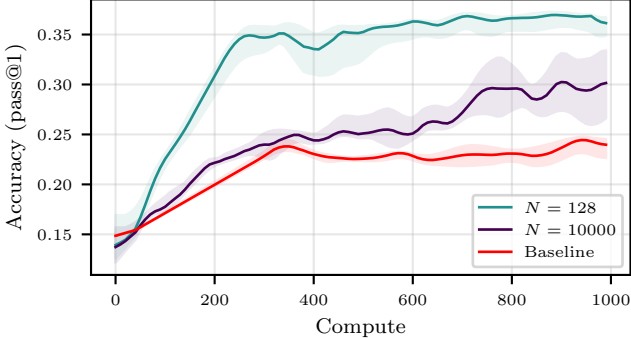

*Figure 16.* **Accuracy with respect to Buffer Size for Qwen3-8B on Lean coding tasks.** Test accuracy as a function of compute spent when training Qwen3-8B on miniF2F for $(W, T) = (6, 2)$ and various buffer sizes $N \in \{128, 10000\}$, as well as for a no-buffer baseline. We report the median and IQR over 4 seeds. Compute is normalized so that each weight update costs 0.55 unit for buffer configurations and 1 for the baseline.

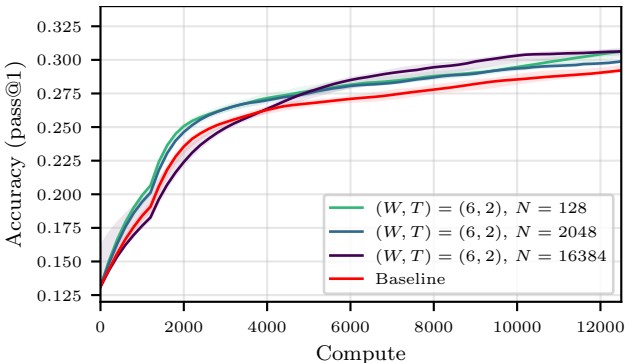

*Figure 17.* **Accuracy with respect to Buffer Size for Llama 3.2 3B.** Test accuracy as a function of compute spent when training Llama 3.2 3B on OpenR1-Math-220k for $(W, T) = (6, 2)$ and various buffer sizes $N \in \{128, 2048, 16384\}$, as well as for a no-buffer baseline. We report the median and IQR over 4 seeds. Compute is normalized so that each weight update costs 0.58 unit for buffer configurations and 1 for the baseline.

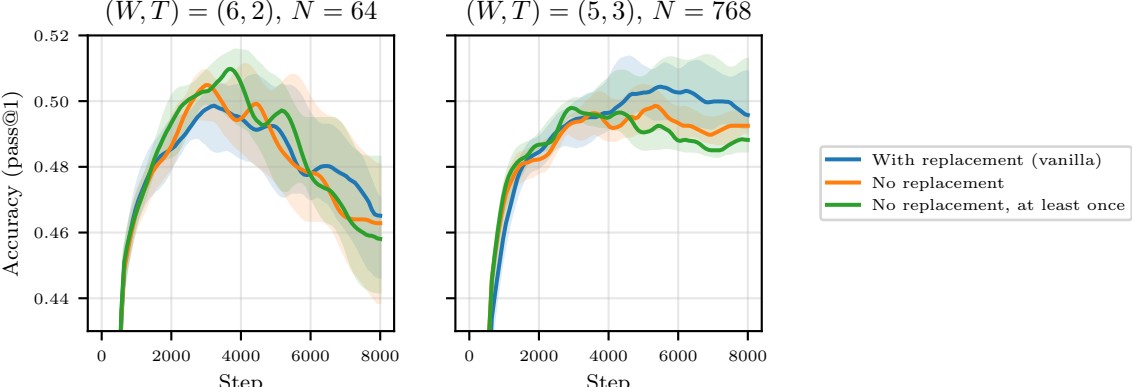

*Figure 18.* We compared our standard buffer implementation, in which the buffer is sampled uniformly by the trainer GPUs ("vanilla"), with two variants: one in which the sampling is done uniformly without replacement ("No replacement"), and one in which samples that have never been used are sampled in priority, after what the remainder of the batch is filled without replacement ("No replacement, at least once"). We did not find any strong signal, as exemplified by these two representative buffer configurations (with which we trained Qwen3-0.6B on OpenR1-Math-220k).

