# OpenReview forum: "Efficient RL Training for LLMs with Experience Replay"
_ICML.cc/2026/Conference — ICML 2026 regular_

### Official Review · Reviewer_F2o1 · 2026-02-19

**Soundness:** 2
**Presentation:** 3
**Significance:** 3
**Originality:** 3
**Overall Recommendation:** 4
**Confidence:** 3

**Summary:**

This work provides the first rigorous mathematical modeling of the trade-off between data staleness and computational efficiency within an asynchronous training architecture. Rather than inventing complex black-box mechanisms, the paper leverages solid theoretical derivations and a reconstruction of cost metrics—using 'Compute' instead of 'Steps' as the evaluation benchmark—to demonstrate that a fundamental Replay Buffer, when properly calibrated, can reduce post-training compute requirements by 40% without compromising, and even potentially enhancing, model diversity.

**Compliance With Llm Reviewing Policy:**

Affirmed.

**Final Justification:**

All of my concerns have been addressed.

**Key Questions For Authors:**

1. Does the "implicit regularization" of experience replay in slowing entropy decay fundamentally differ from the explicit KL penalty in on-policy algorithms (e.g., GRPO) by preserving high-entropy historical states, thereby providing "temporal exploration benefits" and preventing premature convergence to local optima that a static KL constraint cannot achieve?

2. While the paper formalizes the optimality of a FIFO buffer within the proposed framework, is this truly a global optimum, or merely a local optimum reached by balancing system complexity against performance ?

3. In open-ended RLHF scenarios where rewards are derived from continuous and inherently noisy reward models (RM), does ‘staleness-induced variance’ amplify the existing RM noise, and would this interaction lead to a significant shift in the theoretically optimal W/T ratio and buffer size compared to the deterministic reward settings studied in this paper?

**Strengths And Weaknesses:**

## Strengths
1. It provides a better theoretical starting point, rather than a heuristic mechanism.
2. The implicit regularization provided by the replay buffer is an interesting conclusion.
3. Experiment is sufficient.

## Weaknesses
1. The assumption of a linear relationship between data staleness and gradient bias is heuristic and fails to account for the complex, non-linear, or even exponential bias growth and reward clipping effects typical of large-scale LLM training environments (RLHF setting).

2. The empirical validation is overly narrow in scope, limited to relatively small model scales (0.6B, 7B) and specific mathematical reasoning tasks, leaving the framework’s efficacy unproven in high-stakes scenarios such as coding, preference-based RLHF, or at the frontier scales where generation costs are most prohibitive.

3. The baseline comparison is insufficiently rigorous, as it primarily benchmarks against vanilla on-policy RL while overlooking a significant body of recent state-of-the-art methods designed to improve data efficiency, such as RLEP, ReMix, TBA, and EFRame, as well as the latest advantage-based preference optimization frameworks like SAPO and BAPO.

---

> ### Author Rebuttal · Authors · 2026-03-31
>
> We thank the reviewer for their detailed technical questions. We address each concern below, and present new results that substantially broaden the empirical scope. Since submission, we have run experiments with Qwen2.5-32B, Llama3.2-3B, and Qwen3-8B on Lean/MiniF2F (a code generation task involving formal proof synthesis) (see responses to the other reviewers for some of the experimental results).
>
> **Linear staleness assumption.**
>
> The linear staleness model could arise from a Taylor expansion under a boundedness assumption on the related Hessian, which we acknowledge is a strong modeling choice. Different assumptions on the higher-order terms would modify the precise formulas (e.g., in Theorem 4.5) but would preserve the same qualitative trade-off between staleness and diversity. The consistency between our theoretical predictions and experimental results -- across multiple models, tasks (math reasoning, code generation), and buffer sizes -- suggests that the theory captures the dominant effect well enough to provide actionable guidance. Relaxing this assumption is a natural direction for future theoretical work.
>
>
> **Narrow empirical validation.**
>
> Since submission, we have run experiments on Qwen3-8B with Lean/MiniF2F (code generation involving formal proof synthesis), Qwen2.5-32B, and Llama3.2-3B. Key results for Lean/MiniF2F (pass@1, 4 seeds):
>
> | step | 0 | 200 | 280 | 480 |
> |---|---|---|---|---|
> | buffer size 128 | 12.4 $\pm$ 0.4% | 32.4 $\pm$ 5.4% | 37.2 $\pm$ 0.4% | 36.9 $\pm$ 0.7% |
> | no buffer | 12.7 $\pm$ 0.3% | 21.6 $\pm$ 1.8% | 25.7 $\pm$ 4.2% | 35.5 $\pm$ 4.2% |
>
> Buffer-128 reaches near-optimal performance ~3x faster with dramatically lower cross-seed variance. Our results now span model families (Qwen2.5, Qwen3, Llama3.2), scales (0.6B to 32B), and task domains (math reasoning, code generation involving formal proof synthesis).
>
> **Baseline comparison.**
>
> RLEP, ReMix, TBA, and EFRame are either concurrent work or address different aspects of data efficiency (data selection, curriculum design). Our work specifically studies replay buffer design within the standard GRPO pipeline. SAPO and BAPO are preference optimization methods, orthogonal to experience replay. We compare against the most natural baseline: the standard on-policy pipeline, which is the dominant paradigm. Our contribution is demonstrating that a simple modification to this standard pipeline yields significant and consistent gains.
>
>
>
> **Implicit regularization vs. KL penalty.**
>
> Yes, these mechanisms differ in exactly the way you describe: the KL penalty enforces proximity to a fixed reference policy, while replay preserves temporal diversity by training on samples from slightly earlier policies. As you note, this naturally maintains higher-entropy states and provides exploration benefits that a static constraint cannot achieve. The two mechanisms are complementary and could in principle be combined.
>
>
>
> **FIFO optimality.**
>
> As you pointed out, Theorem 4.5 establishes FIFO optimality within our theoretical framework, under the stated assumptions. We view it as an optimality result within a specific (but natural) class of buffer management strategies, rather than a global claim. Indeed, we already explore a variant -- positive-bias sampling -- which departs from pure uniform sampling and can offer benefits in certain regimes, and more sophisticated strategies could offer further improvements.
>
> **RM noise and staleness interaction.**
>
> Within our theoretical framework, RM noise would primarily increase the value of the noise bound $\kappa$ in Assumption 4.2 without changing the functional form of the optimal solution. This means the optimal buffer size may shift downward to limit compounding of staleness and reward noise, but the qualitative trade-off structure -- and the benefit of moderate replay -- is preserved. We note that alternative noise hypotheses (e.g., state-dependent or policy-dependent RM noise) could alter this picture, but should not change the high-level picture.

---

> > ### Author Rebuttal · Reviewer_F2o1 · 2026-04-03
> >
> > I will maintain my score. Thank you for your reply.

---

> > > ### Author Response · Authors · 2026-04-07
> > >
> > > We thank the reviewer for their helpful feedback.

---

### Official Review · Reviewer_JQoc · 2026-02-27

**Soundness:** 3
**Presentation:** 3
**Significance:** 3
**Originality:** 3
**Overall Recommendation:** 4
**Confidence:** 3

**Summary:**

This paper challenges the prevailing "generate-then-discard" paradigm in LLM reinforcement learning (RL) by systematically investigating the utility of experience replay— a foundational technique in classical RL that has been largely overlooked for LLM post-training. The work formalizes the trade-off between data staleness (off-policiness), sample diversity, and computational efficiency, providing both theoretical bounds and empirical validation for optimal replay buffer design. By demonstrating that a well-configured replay buffer can reduce inference compute by up to 40% while maintaining or even improving model performance and policy entropy, this paper makes a practical and impactful contribution to scalable LLM RL. The theoretical analysis is rigorous, the experiments are comprehensive across model scales (Qwen3-0.6B, Qwen2.5-7B) and tasks (mathematical reasoning), and the findings offer actionable guidelines for reducing the prohibitive computational cost of LLM RL training.

**Compliance With Llm Reviewing Policy:**

Affirmed.

**Final Justification:**

After considering both the paper and the authors’ rebuttal, I maintain a slightly positive overall assessment.The paper addresses an interesting and relevant problem, and the proposed approach is reasonably motivated with encouraging empirical results. I find the general direction meaningful, and the work has potential value for the community. The paper is also generally clear and well structured.My initial concerns mainly related to the strength of the empirical evidence and the extent to which the claims are supported. The rebuttal was helpful in clarifying several aspects of the method and evaluation, and I appreciate the authors’ thoughtful responses.Overall, despite these limitations, I view the paper as a solid contribution with reasonable merit. The rebuttal improved clarity, and on balance I am comfortable supporting acceptance.

**Key Questions For Authors:**

The paper validates experience replay on mid-sized models (up to Qwen2.5-7B) but not on larger frontier models (e.g., 70B+ parameters). Could you discuss how compute savings and performance trade-offs scale with model size, and what modifications (if any) would be needed to deploy replay buffers for extremely large LLMs?
While uniform and positive-bias sampling are explored, the paper does not compare to advanced replay strategies (e.g., prioritized experience replay based on TD errors, hindsight experience replay). Could you contextualize the performance of these strategies relative to the proposed methods, and discuss their trade-offs in LLM RL?
The paper quantifies staleness-induced variance but does not explore complementary techniques (e.g., importance sampling corrections) to mitigate staleness. How would these techniques extend the utility of replay buffers, and what are the associated computational costs?
The experiments are limited to mathematical reasoning tasks. How do you expect experience replay to perform on other LLM RL tasks (e.g., code generation, dialogue), and what task-specific adjustments would be needed to balance diversity and staleness?
The paper links buffer size to local vs. global diversity, but the distinction could be more intuitive. Could you provide concrete examples or empirical analysis to clarify how buffer size impacts diversity and downstream performance?

**Limitations:**

No. The authors have not adequately discussed the limitations and potential negative societal impact of their work. Key limitations include:
Limited evaluation on larger frontier models: While the paper validates results on mid-size models (up to Qwen2.5-7B), it does not explore larger frontier models (e.g., 70B+ parameters). This limits claims of scalability to the most compute-constrained large-model RL settings.
Lack of comparison to advanced replay strategies: The paper focuses on uniform sampling and simple positive-bias sampling but does not compare to more advanced strategies from classical RL (e.g., prioritized experience replay based on TD errors, hindsight experience replay). This limits the contextualization of the proposed approach relative to the broader replay literature.
Insufficient analysis of task generalization: The experiments are limited to mathematical reasoning tasks. Validating the approach on other LLM RL tasks (e.g., code generation, dialogue) would demonstrate broader applicability.
Limited discussion of staleness mitigation: While the paper quantifies staleness and its trade-offs, it does not explore complementary techniques (e.g., importance sampling corrections for replay data) that could further extend buffer utility. This restricts the assessment of replay buffers’ maximum potential.
Minor clarity in theoretical presentation: Some aspects of the theoretical analysis (e.g., the link between buffer size and local vs. global diversity) could be more intuitively explained in the main text, particularly for readers less familiar with stochastic optimization in LLM RL.

**Strengths And Weaknesses:**

Strengths:
Novel Challenge to Dominant Paradigm: The paper courageously questions the consensus that strict on-policy data is essential for high-performance LLM RL, highlighting the inefficiency of discarding trajectories after a single use. This reintroduction of experience replay to LLM training fills a critical gap in scalable RL for large models.
Rigorous Theoretical Foundation: The work provides a comprehensive mathematical framework that quantifies the three-way trade-off between staleness-induced variance, sample diversity, and compute cost. The derivation of optimal buffer size and replay ratio (Theorem 4.5) offers principled guidance for practitioners, moving beyond heuristic-based buffer designs in prior related work.
Comprehensive Empirical Validation: The experiments are thorough and well-designed, covering key hyperparameters (buffer size, inference worker-trainer ratio
W/T
), sampling strategies (uniform, positive-bias), and loss functions (GRPO, AsymRE). Results consistently show that replay buffers not only reduce compute but also stabilize training, preserve policy entropy, and sometimes enhance peak accuracy—addressing critical pain points in LLM RL.
Practical Utility and Scalability: The proposed replay buffer is lightweight, easy to implement with minimal modifications to existing asynchronous RL pipelines, and incurs negligible overhead. The compute savings (up to 40%) are substantial and directly address the high inference costs that dominate LLM RL budgets, making it highly relevant for real-world deployment.
Additional Stabilization Benefits: Beyond compute efficiency, the paper uncovers a valuable side effect: replay buffers act as a regularizer, preventing training crashes and preserving output diversity (improving pass@k metrics). This aligns with the need for robust, generalizable LLM reasoning.
Weaknesses & Areas for Improvement:
Limited Evaluation on Larger Frontier Models: While the paper validates results on mid-size models (up to Qwen2.5-7B), it does not explore larger frontier models (e.g., 70B+ parameters). Given that compute constraints are more severe for large models, extending experiments to this scale would strengthen claims of scalability.
Lack of Comparison to Advanced Replay Strategies: The paper focuses on uniform sampling and simple positive-bias sampling but does not compare to more advanced strategies from classical RL (e.g., prioritized experience replay based on TD errors, hindsight experience replay) or recent LLM-specific replay methods. Such comparisons would clarify the relative effectiveness of the proposed approach.
Insufficient Analysis of Task Generalization: The experiments are limited to mathematical reasoning tasks. Validating the approach on other LLM RL tasks (e.g., code generation with more complex benchmarks, logical reasoning, dialogue) would demonstrate broader applicability.
Limited Discussion of Staleness Mitigation: While the paper quantifies staleness and its trade-offs, it does not explore complementary techniques to mitigate staleness (e.g., importance sampling corrections for replay data) that could further extend buffer utility.
Minor Clarity in Theoretical Presentation: Some aspects of the theoretical analysis (e.g., the link between buffer size and local vs. global diversity) could be more intuitively explained in the main text, particularly for readers less familiar with stochastic optimization.

---

> ### Author Rebuttal · Authors · 2026-03-31
>
> We thank the reviewer for their constructive evaluation. Since submission, we have run experiments with Qwen2.5-32B, Llama3.2-3B, and Qwen3-8B on Lean/MiniF2F (a code generation task involving formal proof synthesis) (see the responses to other reviewers for some of the experimental results), directly addressing the concerns about scale and task generalization.
>
>
> **Larger frontier models.**
>
> We have run additional experiments with Qwen2.5-32B, confirming that replay buffers provide compute savings at larger scale. While frontier-scale experiments (70B+) are beyond our current compute budget, the consistency of our results across 0.6B, 3B, 7B/8B, and 32B models, combined with our scale-agnostic theoretical framework, gives us confidence that the findings extend further. The compute savings may in fact be more pronounced at larger scale, as generation costs dominate training costs even more.
> We do not expect any major modification to our buffer implementation to be needed at larger scales, beyond having to communicate sampled trajectories among nodes in highly-parallelised setups.
>
> **Task generalization.**
>
> We trained Qwen3-8B on Lean/MiniF2F, a benchmark where the model must generate formal proofs in the Lean programming language, i.e. a code generation task.
> Results (pass@1, 4 seeds each, each step with a buffer costs ~73% of a baseline step):
>
> | step | 0 | 200 | 280 | 480 |
> |---|---|---|---|---|
> | buffer size 128 | 12.4 $\pm$ 0.4% | 32.4 $\pm$ 5.4% | 37.2 $\pm$ 0.4% | 36.9 $\pm$ 0.7% |
> | buffer size 10K | 12.8 $\pm$ 0.2% | 22.1 $\pm$ 1.5% | 24.3 $\pm$ 3.0% | 30.0 $\pm$ 5.4% |
> | no buffer | 12.7 $\pm$ 0.3% | 21.6 $\pm$ 1.8% | 25.7 $\pm$ 4.2% | 35.5 $\pm$ 4.2% |
>
> Buffer-128 reaches near-optimal performance (~37.2%) by step 280 with remarkably low cross-seed variance ($\pm$ 0.4%), while no-buffer requires ~480 steps and has much higher variance ($\pm$ 4.2%). The 10K buffer shows intermediate early performance but more staleness at later steps, consistent with our theoretical predictions about optimal buffer sizing. These results, on a different model family and task domain, strongly support the generality of our approach.
>
>
> Regarding task-specific adjustments, we expect experience replay to perform similarly on tasks for which the average success rate and trajectory length is similar. On the other hand, we expect modified sampling strategies (e.g. upsampling successful rollouts) to be more promising on extremely hard tasks.
>
> **Advanced replay strategies and staleness mitigation.**
>
> The focus on uniform buffers is deliberate: we wanted to focus on simple implementations that can be incorporated into production pipelines with minimal changes. Note also that importance correction already appears in the GRPO loss that we consider in most of our experiments.
> More sophisticated sampling techniques are natural extensions that we discuss as future work.
>
> **Theoretical clarity.**
>
> We will improve the exposition of the link between buffer size and local vs. global diversity in the camera-ready version. For instance, we will clarify that a small buffer promotes local diversity (samples from nearby policy iterates, reducing staleness but limiting exploration), while a large buffer increases global diversity (samples from more distant policies, introducing broader coverage but also more staleness). We will add concrete examples to make this intuition more accessible.
>
> **Limitations and societal impact.**
>
> Regarding societal impact: our understanding, based on the conference's guidance, was that work contributing to the general progress of ML did not require highlighting potential societal consequences. We are happy to provide a more specific discussion if desired. Regarding scale limitations: we state in the conclusion that further work is needed at larger scales; we can further highlight this. Regarding advanced strategies: we mention that the Pareto frontier can be pushed further with more sophisticated sampling and off-policy corrections, and will emphasize this limitation more clearly. Regarding task generalization: as discussed above, our new Lean/MiniF2F results demonstrate generalization to coding/theorem-proving tasks.

---

> > ### Author Rebuttal · Reviewer_JQoc · 2026-04-02
> >
> > The authors have comprehensively addressed all my concerns. They conducted additional large-scale experiments on frontier models (Qwen2.5-32B, Llama3.2-3B, Qwen3-8B) and a new formal proof synthesis task on Lean/MiniF2F, which strongly validates the scalability and task generalization of their replay buffer approach. The new results clearly demonstrate the compute savings of the method across model sizes, supported by a solid theoretical framework. They also clarified the theoretical link between buffer size and diversity/staleness, committed to improving the exposition in the camera-ready version, and fully addressed all other points including advanced replay strategies, societal impact, and scalability limitations. All my concerns are fully resolved, and I recommend adjusting my score accordingly to accept the paper.

---

> > > ### Author Response · Authors · 2026-04-07
> > >
> > > We thank the reviewer for their helpful feedback and are pleased that our rebuttal addresses their concerns.

---

### Official Review · Reviewer_j1jw · 2026-03-10

**Soundness:** 4
**Presentation:** 4
**Significance:** 3
**Originality:** 3
**Overall Recommendation:** 5
**Confidence:** 5

**Summary:**

This paper explores the efficiency gains that can be had by using experience replay for training language models. As opposed to proposing *new* techniques, this paper strictly analyses the effect that replay buffers can have on training efficiency, stability and performance. Broadly, this analysis focuses on a blend of off-policiness of the data the buffer, the efficiency of the training setup and the diversity of samples (in the buffer, and that are sampled during training from the buffer). The findings are generally very positive for training using replay buffers, with some caveats around the design decisions of the buffers (e.g., how large the buffer is, which affects both the local and global diversity).

**Compliance With Llm Reviewing Policy:**

Affirmed.

**Final Justification:**

This is a solid paper that I feel deserves to be in ICML. I am judging this principally from experimental results, but feel they are sufficient for acceptance even had theory not been included.

I do not give it a 6 as I don't think it has a sufficient 'wow' factor - it is just well done science.

**Key Questions For Authors:**

- Given the intuition that local diversity seems more important than global diversity, one would expect that sampling without replacement would lead to stability improvements. Figure 15 seems directly in opposition to what I would have expected. Why do you think this is the case?
- For pass@k metrics: what value is actually being plotted here? It says 'best accuracy' - is this the best *test* accuracy seen, or the test accuracy corresponding to the best *validation* (or, I guess, train) checkpoint. The latter is significantly more principled than the former, which I would hold some reservations about.
- In Caption 3, it says 'over more than 4 seeds' - at another point in the paper it said '4 random seeds'. How many seeds was it?
- In Figure 8/9, are these learning rates plotted for no replay buffer? Do you think the optimal learning rate would shift for different buffer sizes? It would be interesting (separate to this) to understand how the compute savings could be used to scale up hyperparameter tuning given a larger 'experimental' compute budget.
- Why do you think that most of the field has converged to strictly close-to-on-policy algorithms thus far? Nothing fancy is being applied here, so I'm just curious (personally) why replay buffers work so well here yet have been avoided/neglected by the community until now.

**Limitations:**

There is good discussion of limitations of their study, including results which do not offer significance (I would prefer any paper, but especially an analytical one, to report all results unemotionally rather than try to hide their results.) While these are delegated to the appendix, they are clearly raised in the main body of the paper. The authors also acknowledge that these findings may not transfer to much larger language models, and that further analysis would be an open question.

I think widening the analysis to more than one class of LLMs (i.e., I know this used two separately versioned QWEN models, but it would be good to see if this analysis also held for Llama or Deepseek LLMs too.)

Proposed future work all seems reasonable.

**Strengths And Weaknesses:**

This is a good, clear, and well-written paper. While I will go into a more granular breakdown below, I will say that I recommend acceptance of this work. It's not *ground-breaking* in anything it suggests, but it provides a strong rigourous analysis of the merits and design decisions one should consider when looking to train LLMs with experience replay. Furthermore, while (again) quite intuitive, the formalisation of the computational efficiency of training with and without a buffer is valuable. Clearly, this is a strong reference for the wider community and I believe would be a good contribution to the ICML community. I recommend 5 rather than 6 purely due to the difference between 'high' and 'exceptional' impact.

Strengths:
- This paper answers an interesting and timely question: whether replay buffers can serve a purpose in the LLM-era. I liked the framing of the work in the context of contemporary RL for LLM training (and its focus on being as *on-policy* as possible) while contrasting that to off-policy deep RL. As a (principally) RL researcher, having some additional discussion of off-policy RL, methods applied to scaling the replay ratio etc. would be good, but I do not think it is essential nor should be held against the work.
- The work was well motivated throughout the introduction
- The limitations of the analysis are broached in an upfront way.
- The analysis is carried out in an intuitive way - empirical results based on a standard asynchronous setup and mathematical analysis using a synchronous structure to help intuition.
- I found most of the theory quite intuitive, bar below.
- I liked that most claims were justified in three ways: mathematically, intuitively and empirically.


Weaknesses:
- It feels like there is some juxtaposition between 'Experience Replay in RL' and 'Replay Buffers for LLMs'. On line 90, it says that 'modern LLM reasoning piplines [...] have largely defaulted to on-policy training', but line 96 introduces a number of works in which this is not the case.
- I apprecaite not wanting to redefine terminology, but the caption for figure 1 discusses $W,T$ without definition.
- I'm a bit unclear why PPO is cited for a claim about discarding data when using it only *once*, when the entire objective of PPO is to make multiple updates from the same data without going excessively off-policy. I presume this is trying to contrast with more off-policy methods, but this claim needs to be tightened.
- Theories 4.4 and 4.5 became quite confusing quite quickly - there are some large logical jumps in the paper. I'm not sure if it can really be rectified - referring to the proofs is obviously valid and common, but I still thought I would raise this.
- I appreciate these experiments can get expensive quickly, but it would have been good to include a model from a different family to QWEN too; ensuring that findings transfer not just to models of different size, but also those that have been pre-trained with different recipes.

---

> ### Author Rebuttal · Authors · 2026-03-31
>
> We sincerely thank the reviewer for their positive and thorough evaluation. We have conducted additional experiments with Qwen3-8B on Lean/MiniF2F (a code generation task involving formal proof synthesis), Llama3.2-3B, and Qwen2.5-32B to further validate our findings (see the responses to other reviewers for complete experimental results).
>
> **"It feels like there [...] a number of works in which this is not the case."**
>
> Thank you for pointing this out.
> The dominant paradigm in SOTA pipelines and most of the literature is to be as on-policy as possible, though a few papers have explored variants of experience replay. These efforts remain marginal. We will rephrase this paragraph to make the contrast between mainstream and exploratory approaches clearer.
>
> **"I appreciate [...] without definition."**
>
> Thanks, we will fix it.
>
> **"I'm a bit unclear why PO [...] but this claim needs to be tightened."**
>
> You are right -- what we had in mind were "PPO-like objective functions" such as GRPO, which are typically applied in an on-policy fashion, but the current phrasing is misleading. We will change it.
>
> **"I appreciate [...] have been pre-trained with different recipes."**
>
> We agree. We have run additional experiments with Llama3.2-3B, confirming that replay buffers provide consistent benefits across model families. Results (pass@1, 4 seeds each, each step with a buffer costs ~57% of a baseline step):
>
> | step | 0 | 1000 | 2000 | 3000 | 4000 | 5000 | 6000 | 7000 | 8000 |
>   |---|---|---|---|---|---|---|---|---|---|
>   | buffer size 128 | 13.1 $\pm$ 0.1% | 21.8 $\pm$ 0.2% | 24.9 $\pm$ 0.4% | 25.7 $\pm$ 0.3% | 26.2 $\pm$ 0.2% | 26.9 $\pm$ 0.2% | 27.1 $\pm$ 0.4% | 27.6 $\pm$ 0.6% | 27.9 $\pm$ 0.9% |
>   | buffer size 2048 | 13.2 $\pm$ 0.1% | 21.3 $\pm$ 0.4% | 24.1 $\pm$ 0.2% | 25.6 $\pm$ 0.2% | 26.4 $\pm$ 0.2% | 26.7 $\pm$ 0.3% | 27.1 $\pm$ 0.3% | 27.3 $\pm$ 0.2% | 27.5 $\pm$ 0.1% |
>   | buffer size 16384 | 13.1 $\pm$ 0.0% | 18.7 $\pm$ 0.1% | 21.6 $\pm$ 0.4% | 23.5 $\pm$ 0.2% | 25.0 $\pm$ 0.3% | 25.6 $\pm$ 0.1% | 26.5 $\pm$ 0.2% | 27.4 $\pm$ 0.2% | 28.1 $\pm$ 0.2% |
>   | no buffer | 13.2 $\pm$ 0.3% | 22.5 $\pm$ 0.8% | 25.1 $\pm$ 0.5% | 25.8 $\pm$ 0.3% | 26.6 $\pm$ 0.5% | 27.0 $\pm$ 0.2% | 27.2 $\pm$ 0.5% | 27.7 $\pm$ 0.6% | 27.9 $\pm$ 0.8% |
>
> **"Given the intuition that local diversity [...] Why do you think this is the case?"**
>
> This is an insightful question. In supervised learning, sampling without replacement reduces variance through guaranteed coverage. In our RL setting, however, not all buffer samples are equally useful -- their quality varies due to staleness. Sampling without replacement forces the model to train on every sample in the buffer before any can be reused, including the most stale and least informative ones. With replacement, stale samples are sometimes naturally skipped by chance, providing an implicit form of staleness-aware sampling. Additionally, without replacement imposes a rigid epoch structure over the buffer, which can introduce periodic artifacts in training dynamics that interact poorly with the continuously evolving policy.
>
> **"For pass@k metrics: what value is actually being plotted here?"**
>
> It is the best test accuracy seen. While we agree on principle that using validation-selected checkpoints is more principled, this choice is reasonable: (1) the same method is applied to both baseline and buffer experiments, making comparisons fair; (2) at our evaluation granularity (~200 steps), the best step w.r.t. test accuracy can usually be recovered with high confidence from validation accuracy, which is why best test accuracy is the standard metric in recent LLM work.
>
> **"In Caption 3 [...] How many seeds was it?"**
>
> The number of seeds varies between experiments and configurations (due to varying compute costs and noise levels). All results are reported with at least 4 seeds, sometimes up to 12.
>
> **"In Figure 8/9, are these learning rates plotted for no replay buffer?"**
>
> Yes.
>
> **"Do you think the optimal learning rate would shift for different buffer sizes?"**
>
> We expect (and observed in some experiments) the optimal learning rate to be impacted by the replay ratio, as both affect training stability (see Subsection 5.4). We have not studied the connection between learning rate and buffer size at length, but as a larger buffer sometimes has a stabilizing effect, it could allow for larger learning rates.
>
> **"Why do you think [...] been avoided/neglected by the community until now."**
>
> RL for LLMs in its current form has a very short history, and exploring alternative design choices is costly. We believe many elements of the current SOTA pipeline have not been thoroughly analyzed and optimized.

---

> > ### Author Rebuttal · Reviewer_j1jw · 2026-04-02
> >
> > Thank you for taking the time to write such a detailed rebuttal.
> >
> > I didn't really harbour many concerns about the work beforehand, but have found the points you highlighted really interesting. My only hesitation to give this a 6 is that such a score feels like it should be reserved for "groundbreaking" ideas. I'm going to kepe my score of 5, which I think is a fair reflection of the work. This paper seems quite obviously of the standard for acceptance, and is well-implemented. Therefore, **I would highly recommend the AC accepts this work**.
> >
> > I would note: gkwT has raised some issues with the proofs. I am an empirically-minded researcher, and so while my review has principally focused on the experiments, I think the experimental contribution alone would be sufficient for acceptance.

---

> > > ### Author Response · Authors · 2026-04-07
> > >
> > > We are very grateful to the reviewer for their positive reception of our work, as well as for their helpful comments.

---

### Official Review · Reviewer_gkwT · 2026-03-11

**Soundness:** 3
**Presentation:** 3
**Significance:** 2
**Originality:** 2
**Overall Recommendation:** 4
**Confidence:** 3

**Summary:**

This paper challenges the “generate-then-discard” paradigm in LLM RL post-training. Since inference cost can exceed 80% of the training budget, the authors incorporate a replay buffer into asynchronous pipelines, reusing trajectories multiple times at the cost of increased staleness and reduced diversity. They formalize this as a trade-off between staleness-induced variance $\bar{\sigma}^2$, sample-iterate coupling $\rho$, and inference-to-training compute imbalance $\mu$, deriving optimal buffer size $N/R$ and replay ratio $B/R$. Empirically on Qwen2.5-7B, a well-configured buffer reduces compute by up to 40% while maintaining or improving accuracy.## . Strengths

- **Clearly scoped contribution:** not beating SOTA accuracy, but maximizing accuracy per unit of compute. The 40% compute reduction is practically meaningful.
- The trade-off structure aligns qualitatively with empirical results, suggesting the framework captures the right inductive biases.

---

## 3. Weaknesses

- Main experiments are on Qwen2.5-7B. Generalization to frontier-scale models is unclear, especially since $\mu$ changes with model size and hardware.
- The theoretical contribution is modest: the proof largely follows standard non-convex SGD analysis, and the genuinely novel parts — handling bias and inter-sample correlation — are resolved by Assumptions 4.2 and 4.3, whose validity in the LLM setting is neither proven nor empirically verified (see Q2).ssss

**Compliance With Llm Reviewing Policy:**

Affirmed.

**Final Justification:**

# Strength

1.  The question is well-motivated: the LLM-RL community has defaulted to on-policy methods without rigorous justification. A principled analysis of replay buffers fills a genuine gap.

2.  The rebuttal substantially strengthened the experimental coverage. The paper now spans three model families (Qwen2.5, Qwen3, Llama3.2), four scales (0.6B–32B), and two distinct task domains (MATH/AIME and Lean/MiniF2F)

# Weaknesses

1. Assumption 4.3 remains empirically unverified beyond a single point estimate ($\kappa \sim 100$), and the gap between the generality of the assumptions and the limited empirical validation is not closed. Since the theoretical analysis is advertised as one of the  core contributions, and these assumptions are critical to deriving the main results, this weakness is difficult to overlook.

# Conclusion

While the weakness is notable, the breadth of empirical validation, spanning multiple model families, scales, and task domains, is convincing, and the practical implications are immediate and broadly relevant to the community. The paper is clearly written, honest about its limitations, and addresses a timely question that has received insufficient scrutiny. On balance, the strengths of the empirical contribution and the utility of the theoretical framework as a qualitative guide outweigh the gap in theoretical validation, and I recommend **weak accept.**

**Key Questions For Authors:**

**Q1. Are the key assumptions doing too much work in the proof?**

The proof largely follows standard SGD analysis. My opinion is that the genuinely novel theoretical difficulty of Thorem 4.4 -- 4.5, handling bias and correlation introduced by the buffer, seems to be resolved almost entirely by Assumptions 4.2 and 4.3. Assumption 4.2 requires

$
\|\pi\_{\theta\_{t-t\_i}}(\cdot|\mathcal{F}\_t) - \pi\_{\theta\_t}\| \leq \kappa\_0 \|\pi\_{\theta\_{t-t\_i}} - \pi\_{\theta\_t}\|,
$

which relates conditional and unconditional distribution distances in a way that does not hold in general. Assumption 4.3 linearizes a complex inter-sample dependency structure via $\text{corr}(\varepsilon\_{t,i}, \varepsilon\_{t,j}) \leq \rho|t\_i - t\_j|/N$.
Although the authors justified these assumptions in the technical appendix, I belive the following: are there any empirical checks — e.g., directly measuring $\|\mathbb{E}[\varepsilon\_{t,i}|\mathcal{F}\_t]\|$ or inter-sample correlations during training — that validate these assumptions in the LLM setting?

**Q2. Steady-state assumption.**

The theoretical analysis implicitly assumes the buffer is full (NN
N samples), which requires at least $N/R$ warm-up steps. However, early training — when the policy changes most rapidly — operates with a partially filled buffer, where the actual staleness distribution deviates significantly from the assumed uniform $[0,N/R]$. How does this transient phase affect the validity of the theoretical conclusions, and does it have any practical impact on the recommended buffer design?

**Limitations:**

yes

**Strengths And Weaknesses:**

## . Strengths

- **Clearly scoped contribution:** not beating SOTA accuracy, but maximizing accuracy per unit of compute. The 40% compute reduction is practically meaningful.

- The trade-off structure aligns qualitatively with empirical results, suggesting the framework captures the right inductive biases.

---

## 3. Weaknesses

- Main experiments are on Qwen2.5-7B. Generalization to frontier-scale models is unclear, especially since $\mu$ changes with model size and hardware.
- The theoretical contribution is modest: the proof largely follows standard non-convex SGD analysis, and the genuinely novel parts — handling bias and inter-sample correlation — are resolved by Assumptions 4.2 and 4.3, whose validity in the LLM setting is neither proven nor empirically verified (see Q2).

---

> ### Author Rebuttal · Authors · 2026-03-31
>
> We thank the reviewer for their thorough and constructive evaluation. Since submission, we have run additional experiments to address the concerns about model scale and task generalization: Qwen2.5-32B and Llama3.2-3B on MATH/AIME, and Qwen3-8B on Lean/MiniF2F (a code generation task involving formal proof synthesis) (see the responses to other reviewers for some of the experimental results).
>
> **"Main experiments [...] especially since $\mu$ changes with model size and hardware."**
>
> We agree that the compute split between generation and training varies with scale, and that extending our analysis to bigger models strengthens our conclusions. We have run experiments with Qwen2.5-32B (pass@1, each step with buffer costs roughly 75% of a step without):
>   | step | 0 | 150 | 300 | 450 | 600 | 750 | 900 | 1050 | 1200 |
>   |---|---|---|---|---|---|---|---|---|---|
>   | no buffer | 35.6% | 36.6% | 37.8% | 38.7% | 41.1% | 40.8% | 42.2% | 42.9% | 43.8% |
>   | buffer size 512 | 35.9% | 36.9% | 37.6% | 39.3% | 41.2% | 41.1% | 42.2% | 43.5% | 43.5% |
>
> confirming that replay buffers remain beneficial at larger scale. We also trained Qwen3-8B on Lean/MiniF2F, a formal theorem proving benchmark that uses a different model family (Qwen3 vs Qwen2.5) and a fundamentally different task domain. On this benchmark, buffer-128 reaches 32.4% pass@1 by step 200 vs 21.6% without a buffer. By step 280, buffer-128 converges to 37.2 $\pm$ 0.4% across all 4 seeds, while no-buffer is still at 25.7 $\pm$ 4.2%. This consistency across seeds, model families, and task domains supports the robustness of our findings. We also ran experiments with Llama3.2-3B, demonstrating generalization across architectures.
>
> We acknowledge that compute constraints limit the scale of experiments we can run within the rebuttal period, but the evidence across three model families (Qwen2.5, Qwen3, Llama3.2), four scales (0.6B, 3B, 7B/8B, 32B), and two task domains provides substantial coverage.
>
> **"Q1. Are the key assumptions doing too much work in the proof?"**
>
> We thank the reviewer for this question. We have conducted preliminary experiments to estimate kappa empirically. We find that kappa ~ 100 seems to be a reasonable estimate, which is consistent with our bound. Furthermore, we observe experimentally that both terms in the bound vanish together, confirming the expected behavior. We are currently running additional experiments to obtain a more detailed characterization of kappa, and will update the paper accordingly.
> Directly measuring the correlation requires estimating joint conditional expectations, making precise empirical evaluation difficult.
>
> **"Q2. Steady-state assumption."**
>
> As the theoretical conclusions relate to asymptotic behaviour, they remain unaffected by what happens in the first K steps for any finite K. The rate of convergence, which is the object of the theorems, does not depend on the transient phase.
>
> In practice, this is not an issue either. While we expect minor degradation in the "quality" of early training, the effect is too small to observe in the trajectories. We typically consider buffer sizes smaller than 4000, which are filled in less than 100 steps with our batch sizes (often less than 20 steps), while runs last more than 5000 steps. Despite early training being more "eventful", the impact of such a short transitional period is small, especially as a linear warm-up is applied to the learning rate (as in our experiments), further reducing the importance of early steps. One could also wait until the buffer is full before the first training step, but our preliminary experiments suggest this does not make a significant difference.

---

> > ### Author Rebuttal · Reviewer_gkwT · 2026-04-02
> >
> > I thank the authors for the thorough rebuttal. The additional experiments across three model families (Qwen2.5, Qwen3, Llama3.2), four scales, and two task domains convincingly address my concerns about generalization, and the response to Q2 (steady-state assumption) is satisfactory.
> >
> > **Regarding Q1:** I appreciate the preliminary empirical estimate of $\kappa \sim 100$ and the observation that both terms in the bound vanish together. However, Assumption 4.3 remains empirically unverified, and the $\kappa$ characterization is still a single point estimate rather than a systematic validation, thus they are still unclear (at least to me). Since the theoretical analysis is presented as a core contribution of the paper — and the key novelty within that analysis hinges precisely on these assumptions — this gap is not easily overlooked.
> >
> > **Conclusion:** I raise my score from 3 to 4, but do not see a path to raising it further due to the unresolved concern in Q1.

---

> > > ### Author Response · Authors · 2026-04-07
> > >
> > > We thank the reviewer for their positive response to our rebuttal and for their helpful feedback.

---

### Decision · Program_Chairs · 2026-04-30

**Decision:**

Accept (regular)

**Comment:**

There is consensus among reviewers that this paper is ready to be accepted to the conference. Citing one of the reviewer's comment, which the SPC also think is a great summary of the paper: while there is not a sufficiently "wow" factor, the paper is well-done science. The authors did great job to empirically demonstrate a practically useful approach, accompanied with a good amount of theoretical explanations. The reviewers also appreciated the honesty of the statements as well as discussion of potential limitations.